# Hydra: Bidirectional State Space Models Through Generalized Matrix Mixers

**Sukjun Hwang**[*,1]**, Aakash Lahoti**[*,1]**, Ratish Puduppully**[†,2]**, Tri Dao**[3]**, and Albert Gu**[1,4]

[1]Machine Learning Department, Carnegie Mellon University
[2]IT University of Copenhagen
[3]Department of Computer Science, Princeton University
[4]Cartesia AI
{sukjunh,alahoti}@cs.cmu.edu, rapu@itu.dk, tri@tridao.me, agu@cs.cmu.edu

## Abstract

A wide array of sequence models are built on a framework modeled after Transformers, comprising alternating sequence mixer and channel mixer layers. This paper studies a unifying *matrix mixer* view of sequence mixers that can be conceptualized as a linear map on the input sequence. This framework encompasses a broad range of well-known sequence models, including the self-attention of Transformers as well as recent strong alternatives such as structured state space models (SSMs), and allows understanding downstream characteristics such as efficiency and expressivity through properties of their structured matrix class. We identify a key axis of matrix parameterizations termed *sequence alignment*, which increases the flexibility and performance of matrix mixers, providing insights into the strong performance of Transformers and recent SSMs such as Mamba. Furthermore, the matrix mixer framework offers a systematic approach to developing sequence mixers with desired properties, allowing us to develop several new sub-quadratic sequence models. In particular, we propose a natural bidirectional extension of the Mamba model (**Hydra**), parameterized as a *quasiseparable matrix mixer*, which demonstrates superior performance over other sequence models including Transformers on non-causal tasks. As a drop-in replacement for attention layers, Hydra outperforms BERT by 0.8 points on the GLUE benchmark and ViT by $2\%$ Top-1 accuracy on ImageNet.

## 1 Introduction

Large-scale pretrained models such as GPT [35], BERT [10], and ViT [11] exhibit state-of-the-art performance across a wide range of tasks in multiple domains, including language and vision. A large number of these pretrained models follow a multi-layer architectural blueprint: a *sequence mixer*, such as Self-Attention[1] [42], aggregates information across the input sequence, followed by a *channel mixer* processes information within each sequence element. Over the years, Attention has been the predominant choice for sequence mixing due to its ability to facilitate direct pairwise interactions between elements of the input sequence in a single step. However, this capability incurs a quadratic cost with respect to sequence length, making the training and deployment of these models prohibitively expensive for longer sequences. Although numerous alternatives have been proposed, designing principled sequence models that match the performance and versatility of attention-based systems remains a substantial challenge.

One general strategy in designing alternative sequence models involves substituting the Attention matrix with different matrix parameterizations as the core sequence mixer. These modifications are motivated by various goals. For instance, simplifying the sequence mixer has led to the development of models such as MLP-Mixer [40], which uses dense matrices, and FNet [23], which utilizes the Discrete

---

[*]Equal Contributions.

[†]Work done at A*STAR, Singapore.

[1]In this paper, Attention [3] exclusively refers to Self-Attention.

38th Conference on Neural Information Processing Systems (NeurIPS 2024).

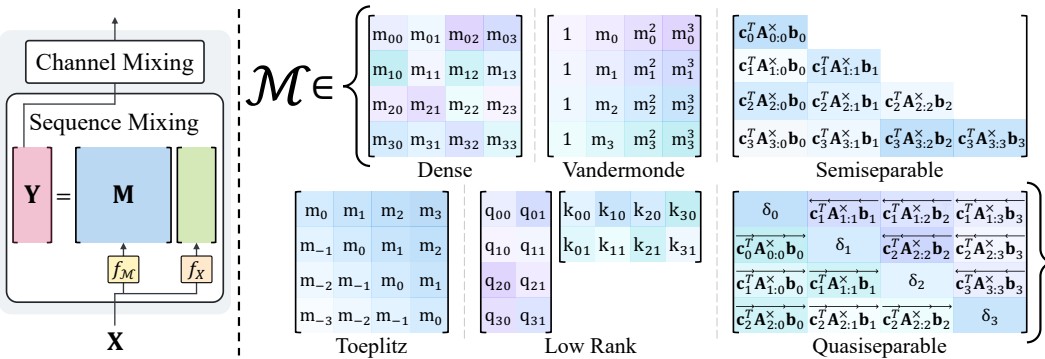

Figure 1: (Left) A schematic of the matrix mixer framework. (Right) An overview of matrix mixer classes: dense, Vandermonde, Toeplitz, low-rank, semiseparable, and quasiseparable.

Fourier Transform matrix. Additionally, incorporating inductive biases such as positional information has resulted in the use of Toeplitz matrices in models like TNN [33]. Enhancing computational efficiency has spurred the creation of low-rank structures in models like Linear Attention (LA) [22] and Linformer [45], as well as Monarch matrices [5] in models such as M2 [13].

Despite the achievements of these approaches, they often lack a systematic analysis of their theoretical foundations and empirical consequences. Moreover, these models typically exhibit lower empirical performance than Attention, which is successfully applied across diverse domains and tasks. In another line of work, structured state space models (SSMs) [17, 18] such as Mamba [16] have been popularized for its attention-like performance with linear-time computational scaling. Furthermore, recent work [6] has shown that SSMs can also be expressed as semiseparable matrices. However, these models are primarily causal as underscored by their recurrent view of computation, limiting their application to auto-regressive settings. Several concurrent efforts to adapt Mamba into a bidirectional model [48, 38] have been made, but these attempts remain ad-hoc.

In this work, we study the **matrix mixer** framework (see Figure 1), which provides a powerful abstraction for enhancing the understanding of sequence models through the analysis of their matrix structures. Specifically, we suggest the following:

(i) **Formalization of matrix mixer sequence models.** We establish the conceptual foundation of the Matrix Mixer sequence models, delineating key axes of matrix parameterizations that influence model characteristics such as efficiency, causality, and expressivity. For instance, we highlight that structured matrix [29] parameterizations underpin efficient sequence models through computationally efficient algorithms (Section 2.1).

(ii) **Introduction of sequence alignment.** Within this framework, we define *sequence alignment* as a novel attribute, which induces sequence models with essential features of *data-dependence* and *extendability*. We introduce *Sequence Aligned Matrices* (SAM) that exhibit these properties, demonstrating their superior performance in downstream tasks compared to non-aligned counterparts (Section 2.2).

(iii) **Exploration of structured matrix parameterizations.** Through the matrix mixer framework, we systematically explore and categorize a broad spectrum of existing sequence models (Section 2.3). Motivated by sequence alignment, we also present new sequence models with underexplored structured matrix configurations, such as Vandermonde and Cauchy matrices (Section 2.4).

Building on our matrix mixer framework, we introduce a novel sequence model **Hydra**, which employs a *quasiseparable* matrix mixer (Section 3). Quasiseparable matrices are a fundamental matrix structure with several important properties, making them ideal for sequence modeling. For example, they generalize both the low-rank matrix mixers found in linear attention models and the semiseparable matrices utilized in state space models. In fact, quasiseparable matrix mixers can be seen as a natural bidirectional extension of semiseparable matrices, addressing a major limitation in state space models by making their use in non-causal settings possible. Additionally, Hydra maintains the strong performance and linear-time computational efficiency of SSMs, thanks to structured matrix multiplication algorithms. Unlike prior attempts to make SSMs bidirectional by ad-hoc methods – typically by combining separate models for forward and backward sequence processing through element-wise addition [15, 48, 38], the Hadamard product [44], or concatenation [44, 12, 38] – the systematic approach of Hydra offers a more coherent and theoretically grounded advancement. Specifically, the definition of quasiseparable mixer matrices generalize both the heuristic bidirectional extensions of SSMs and linear attention, providing a mathematical interpretation of the strong expressivity exhibited by Hydra.

We provide extensive experimental results that substantiate our claims. Our systematic ablation studies control architectural variables to highlight the impact of matrix parameterization. These careful experiments confirm that Sequence Alignment, a property we newly identified in certain matrix mixers, significantly enhances downstream performance. Furthermore, our experimental findings demonstrate that novel sequence models like those using Cauchy matrix mixers match the performance of established matrix mixers such as low-rank. This indicates that the strong performance of low-rank variants is not solely attributable to their matrix structure, but can be matched by other mixers that share similar properties. In addition, we also validate that the bidirectional extension of the Mamba model, implemented through quasiseparable matrix mixers, outperforms naive bidirectional approaches [15, 48, 38, 44, 12]. Importantly, Hydra excels as a performant, general-purpose bidirectional sequence model, as evidenced by its strong performance across diverse domains. It achieves state-of-the-art results on the GLUE benchmark [43] with an average accuracy of $84.3\%$, outperforming BERT [10] by $0.8$ points, and records a Top-1 accuracy of $81.0\%$ on the ImageNet-1K benchmark [9], surpassing ViT [11] by 2.2 points.

We publicly release source code at https://github.com/goombalab/hydra.

## 2 The Matrix Mixer Framework: Bridging Sequence Mixers and Structured Matrices

In Section 2.1, we formally define the matrix mixer framework, conceptualizing sequence mixers as linear maps on input sequences. In Section 2.2, we introduce sequence alignment, a new axis of variation of matrix structures which controls important characteristics of downstream sequence models such as data-dependent parameterizations and extendability. Section 2.3 leverages these definitions to categorize a wide array of previous works based on their matrix mixers, facilitating the understanding of sequence mixers through their matrix structures. Furthermore, in Section 2.4, we propose novel sequence mixers utilizing Vandermonde and Cauchy matrices, demonstrating the flexibility of our framework in systematically designing new sequence mixers.

### 2.1 Formalizing the Matrix Mixer Framework

**Definition 2.1** (*The matrix mixer framework*). *Let $\mathbf{X} \in \mathbb{R}^{L \times C}$ be the input sequence, consisting of $L$ elements, each with $C$ channels. Let $f_X : \mathbb{R}^{L \times C} \to \mathbb{R}^{L \times D}$ be the input preprocessing function that encapsulates common data transformations such as short convolutions, and linear layers. Let $H$ and $P$ be the number of heads and head dimension respectively, such that $HP = D$. Let $\mathcal{M} \subseteq \mathbb{R}^{L \times L}$ represent the underlying class of mixer matrices. For each head $h \in [H]$, let $f_{\mathcal{M}}^h : \mathbb{R}^{L \times C} \times \Theta \to \mathcal{M}$ be the matrix construction function that maps input data to mixer matrices, where $\Theta$ is the space of learnable parameters. We denote $\mathbf{M}^h = f_{\mathcal{M}}^h(\mathbf{X}, \theta)$, when $\mathcal{M}, \theta, \mathbf{X}$ are clear from the context. Then, we define matrix mixing as,*

$$\mathbf{Y}^h = \mathbf{M}^h (f_X(\mathbf{X}))^h,$$

*where $(f_X(\mathbf{X}))^h, \mathbf{Y}^h$ denote the preprocessed input and output slice corresponding to head $h$.*

This definition encapsulates the view that existing sequence mixers can be mathematically represented as $L \times L$ mixer matrices that act on the sequence length (see Figure 1, left). The framework not only incorporates typical data preprocessing steps like projections [22, 39], and short convolutions [16, 6], but it also accommodates data dependent mixer matrices [39, 34]. Furthermore, it is also powerful enough to capture the concept of head structure [42] by equivalently sharing the mixer matrices within a head.

Moreover, Definition 2.1 offers a different lens to conceptualize the computational cost of sequence mixers. If we assume that both the preprocessing function and the matrix construction function are sub-quadratic in the sequence length $L$, which is generally true in practice, then the computational bottleneck lies in the $\mathbf{M} f_X(\mathbf{X})$ operation. For a general matrix $\mathbf{M}$, this multiplication will incur a $O(L^2)$ cost. The only way to mitigate this to restrict $\mathcal{M}$ to being a *structured matrix*, which are known to possess sub-quadratic matrix multiplication algorithms. We refer to such sequence mixers as *structured matrix mixers*. This paves the way to systematically develop new sequence mixers: select an appropriate class of structured matrices from established mathematical literature, devise a data-dependent matrix construction function, and integrate them into the matrix mixer framework.

### 2.2 Sequence Aligned Matrices

Unstructured mixer matrices, also known as dense matrices, lack two key useful properties: 1) these matrix mixers cannot be easily made *data-dependent*, which has been identified as a key property of performant sequence models such as Transformers and Mamba, and 2) they can only be applied to fixed-length sequences, a property we call *extendability*. Due to substantial importance of these

Table 1: Categorization of existing methods as matrix mixers. $L$ denotes input sequence length.

| Matrix Structure $\mathcal{M}$ | Formulation ($m_{ij}$) | Complexity | Sequence Aligned | Method Instantiations |
|---|---|---|---|---|
| Dense | $m_{ij}$ | $O(L^2)$ | | MLP-Mixer [40] |
| Dense (Softmax Attention) | $\mathrm{softmax}_j(\mathbf{q}_i^T\mathbf{k}_j)$ | $O(L^2)$ | ✓ | Transformer [42] |
| Low-Rank (Linear Attention) | $\mathbf{q}_i^T\mathbf{k}_j$ | $O(L)$ | ✓ | Linear Attention [22], Linformer [45] |
| Butterfly | See [7, 5] | $O(L\log L)$ | | Kaleidoscope [8, 7], Monarch [5, 13] |
| Toeplitz (Convolution) | $m_{j-i}$ | $O(L\log L)$ | | S4 [17, 18], H3 [14] TNN [33], CKConv [37] |
| Discrete Fourier Transform | $w^{ij}$ | $O(L\log^2 L)$ | | FNet [23] |
| Vandermonde Cauchy | $\dfrac{(m_i)^j}{\sum_d (q_{id}-k_{jd})^{-1}}$ | $O(L\log^2 L)$ | ✓ | Ours (Section 2.4) |
| Semiseparable | $\mathbf{c}_i^T \mathbf{A}_{i:j}^\times \mathbf{b}_j \mathbb{1}_{\{i\geq j\}}$ (1), (2) | $O(L)$ | ✓ | Mamba (S6 [16], SSD [6]) |
| Quasiseparable | Equation (3) | $O(L)$ | ✓ | Ours (Hydra) (Section 3) |

features for sequence mixers, we formally define the class *Sequence Aligned Matrices* (SAM) to systematically explore matrices that are characterized with both properties.

**Definition 2.2** (*Sequence Aligned Matrices*). *Let $L$ be the sequence length and let $\mathbf{M}\in\mathbb{R}^{L\times L}$ denote a matrix with a parameter set $\mathcal{P}$. Then, we say that $\mathbf{M}$ is a Sequence Aligned Matrix if there exists a partition $\Pi$ of $\hat{\mathcal{P}}\subseteq\mathcal{P}$, and $\hat{\mathcal{P}}\neq\varnothing$, such that for all sets $\mathcal{E}\in\Pi$, there exists a bijective map $f_\mathcal{E}:[L]\to\mathcal{E}$, and, for each $i,j\in[L]$, $i\leq j$, the sub-matrix $\mathbf{M}[i:j+1,i:j+1]$ is composed solely from the parameters in the subset $\cup_{\mathcal{E},i\leq k\leq j}f_\mathcal{E}(k)\subseteq\mathcal{P}$.*

In simpler terms, this implies that each parameter of a SAM is either mapped to a specific element of the sequence or is left data-independent, ensuring that every upper-left sub-matrix upto index $i$ is constructed using only the parameters corresponding sequence segment upto and including index $i$ and/or the data-independent ones.

**Proposition 2.3** (Data Dependency). *Sequence aligned matrices exhibit canonical data-dependent parameterization.*
*Proof.* This property arises from the parameter partition structure guaranteed in the definition. Specifically, for each partition we associate a parametric function that, for any given element $i$ computes the parameter's value by treating the element itself as an input. □

**Proposition 2.4** (Extendability). *Sequence aligned sequence mixers can be extended beyond their trained length.*
*Proof.* This is a direct consequence of Proposition 2.3: the pretrained parametric functions assigned to each partition enable the computation of matrices larger than those encountered during training. □

We identify data dependency – the property induced by SAM – as a key axis of differentiation amongst existing models. Although data-dependency is a popular notion that is widely regarded as being crucial to performance of Attention, it lacked any formal definition in the literature. Consequently, various works have adopted different interpretations of this notion: Hyena [31] implements it via a data-dependent linear operators over the Toeplitz matrix mixer; GSS [26] adds a data-dependent gate post sequence mixing; LA, SSD [22, 6], like Attention, directly map input data to the parameters of the matrix mixer. We adopt the third notion of data dependency, where each parameter is a function of a particular input token, and under this definition it is clear that models like Hydra, SSD, and LA are data-dependent, whereas MLP-Mixer, FNet, S4, S4D, and TNN, only have data-independent parameters.

## 2.3 Prior Sequence Models as (Structured) Matrix Mixers

Using the formalization of the Matrix Mixer framework, we categorize a wide array of previous works – MLP-Mixer [40], Transformer [42], Linear Attention [22], Linformer [45], S4 [18, 17], H3 [14], TNN [33], CKConv [37], FNet [23], Kaleidoscope [8, 7], Monarch [5, 13], and Mamba [16, 6] – as matrix mixers in Table 1. For illustrative purposes, we explicitly show that MLP-Mixer, FNet and LA are matrix mixers (proofs in Appendix B), and leave out normalizing factors for simplicity as follows:

**Proposition 2.5.** *(MLP-Mixer is a matrix mixer). MLP-Mixer employs $f_X$ as an identity function, and its mixer matrix $\mathbf{M}$ has an unstructured parameterization with a single head ($H=1$).*

**Proposition 2.6.** *(LA is a structured matrix mixer with sequence alignment). LA employs $f_X$ as a linear projection $f_X(\mathbf{X},W_V) = \mathbf{X}W_V$, with its low-rank mixer matrix being SAM. The mixer matrix $\mathbf{M}$ has $H$ heads with a structured parameterization, where each $(i,j)$-element $m_{ij}$ is defined as:*

$$m_{ij} = \phi(x_i W_Q)\phi(x_j W_K)^T, \qquad \phi(\cdot) = elu(\cdot)+1, \qquad W_Q, W_K \in \mathbb{R}^{C \times D}.$$

Additionally, it is important to note that not all structured matrix mixers inherently exhibit sequence alignment. For example, consider the matrix mixer formulation $\mathbf{M} = W_Q W_K^T$ where $W_Q, W_K \in \mathbb{R}^{L \times C}$ are trainable parameters. This formulation yields a low-rank matrix mixer, which qualifies as a structured matrix mixer. However, this structure is independent of the input data, directly contradicting the concept of SAM. Another illustrative example is FNet:

**Proposition 2.7.** *(FNet is a structured matrix mixer without sequence alignment). FNet employs $f_X$ as an identity function, using the canonical Vandermonde matrix – Discrete Fourier Transform (DFT) – for its mixer matrix. The mixer matrix $\mathbf{M}$ has a single head ($H=1$) with a structured parameterization, where each $(p,q)$-element $m_{pq}$ is defined as $m_{pq} = w^{pq}$, where $w = e^{-\frac{2\pi i}{N}}$.*

Using these definitions, we present a series of matrix mixers, incorporating various mixer matrices both with and without the SAM property, and provide extensive experimental evaluations to investigate the influence of SAM on the expressivity of sequence mixers in Section 4.1.1.

### 2.4 The Matrix Mixer Framework as a Creative Toolbox

Armed with this framework, we illustrate how more classes of performant sequence models can be derived. In particular, we introduce Vandermonde and Cauchy matrix mixers, chosen because they are: 1) well-known families of structured matrices with sub-quadratic matrix multiplication, hence are efficient as sequence models, and 2) we show that they have SAM parameterizations. In Section 4.1.1, we validate that these properties are enough to create efficient and strong sequence models, e.g. our Cauchy matrix mixer is on par with LA. This validates the central theme of this paper that developing sequence models can be reduced to choosing structured matrix classes with target properties. We assume an input sequence $\mathbf{X} \in \mathbb{R}^{L \times C}$ is given, and utilize the prevalent query-key concept that we employ $\mathbf{Q} = \mathbf{X}W_Q$, $\mathbf{K} = \mathbf{X}W_K$ where $W_Q, W_K \in \mathbb{R}^{C \times D}$. For simplicity, we show the single-headed ($H=1$) case, where $D=P$. Further details including the implementations of multi-headed extensions are in Appendix E.

**Sequence aligned Vandermonde mixer.** In contrast to FNet, which employs a fixed-parameter Vandermonde mixer based on the Discrete Fourier Transform (DFT), we propose a trainable, and data-dependent Vandermonde-based matrix mixer. Then, we can construct a mixer matrix from a combination of two separate Vandermonde matrices $\mathbf{M}_Q, \mathbf{M}_K$, each generated from $\mathbf{Q}$ and $\mathbf{K}$. Specifically, each $(i,j)$-element of the resulting mixer matrix $\mathbf{M}$ is formulated as $m_{ij} = \sum_d (\cos(2\pi q_{id}^j) - \cos(2\pi k_{jd}^i))$. This formulation utilizes the cosine function – the real part of an imaginary number – which is a well established technique used in SSMs [17, 28] to prevent the potential for excessive values resulting from the powers of elements. Thanks to the associative property of matrix multiplications, our sequence aligned Vandermonde matrix mixer enjoys algorithms with efficient computational complexities.

**Sequence aligned Cauchy mixer.** As indicated by the definition of Cauchy matrices in Table 1, the underlying philosophy of using both the Cauchy and low-rank matrix classes as a sequence mixer is remarkably similar: the relevance of a pair of elements is directly proportional to their associated value – the greater the relevance, the higher value assigned. Specifically, each $(i, j)$-element of the mixer matrix $\mathbf{M}$ is constructed by $m_{ij} = \sum_d 1/(q_{id} - k_{jd} + c)$, where $c$ is a trainable constant that prevents elements leading to excessive values when the denominator $q_{id} - k_{jd}$ approaches zero. To our knowledge, this is the first introduction of a Cauchy-based sequence mixer that achieves performance comparable to Attention matrix mixers.

## 3 Hydra: The Double-Headed Mamba

In this section, we introduce Hydra, a novel sequence-to-sequence model through a bidirectional extension of Mamba. We briefly begin with a preliminary background that SSMs are semiseparable matrix mixers (Section 3.1). Then, we motivate the design choice of Hydra under the domain of the matrix mixer framework. Specifically, we opt for quasiseparable matrices as our matrix mixer, a choice grounded by solid mathematical foundations (Section 3.2). Additionally, we underline the computational benefits and enhanced parameter efficiency afforded by adopting quasiseparable matrices (Section 3.3).

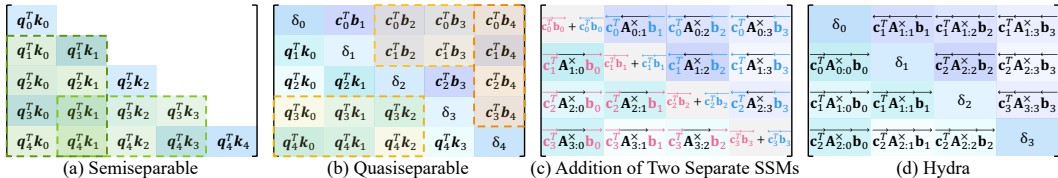

$$\begin{array}{cccc} \text{(a) Semiseparable} & \text{(b) Quasiseparable} & \text{(c) Addition of Two Separate SSMs} & \text{(d) Hydra} \end{array}$$

Figure 2: (a) A semiseparable (SS) matrix. (b) A quasiseparable (QS) matrix. (c) A mixer matrix of addition-based bidirectional SSMs. (d) A QS mixer matrix for Hydra. SS and QS matrices are characterized by rank conditions (Definition 3.1, Definition 3.2). The rank characterization of SS matrices include the diagonals (*e.g.,* green submatrices), whereas that of QS matrices hold for off-diagonal submatrices (*e.g.,* yellow submatrices). Because of the similar rank properties, a naive addition-based bidirectional SSM is provably a QS matrix mixer. Hence, QS matrix mixers generalize this common heuristic for bidirectional SSMs. The freedom in the diagonal values of Hydra leads to a higher expressivity compared to the mixer matrices of the addition-based bidirectional SSMs, where the diagonal values are constrained by the colored vectors.

## 3.1 Background: State Space Models are Semiseparable Matrix Mixers

SSD [6], the latest advancement in the iterations of SSMs, presents a sub-quadratic sequence mixer that attains language modeling performance on-par with Attention. Crucially, SSD underscores that all SSMs are inherently parameterized by semiseparable matrices, which play a pivotal role in their computational efficiency and strong empirical performance. Specifically, the operational essence of selective SSMs such as Mamba – the transformation of an input sequence $\mathbf{X} \in \mathbb{R}^{L \times C} \mapsto \mathbf{Y} \in \mathbb{R}^{L \times C}$ – can be succinctly represented within the matrix mixer framework, as detailed below:

$$
\begin{aligned}
\mathbf{y}_t &= \sum_{s=0}^{t} \mathbf{C}_t^T \mathbf{A}_{t:s}^{\times} \mathbf{B}_s \mathbf{x}_s \\
\mathbf{Y} &= \mathrm{SSM}(\mathbf{A}, \mathbf{B}, \mathbf{C})(\mathbf{X}) = \mathbf{M}\mathbf{X} \\
m_{ij} &= \mathbf{c}_i^T \mathbf{A}_i \cdots \mathbf{A}_{j+1} \mathbf{b}_j
\end{aligned}
\tag{1}
\qquad
\mathbf{A}_{i:j}^{\times} = \begin{cases} \prod_{k=j+1}^{i} \mathbf{A}_k & \text{for } i > j \\ 1 & \text{for } i = j \\ \prod_{k=i}^{j-1} \mathbf{A}_k & \text{for } i < j \end{cases}
\tag{2}
$$

Here, $m_{ij}$ represents an $(i, j)$-element of the mixer matrix $\mathbf{M} \in \mathbb{R}^{L \times L}$, with each matrix $\mathbf{A}_i \in \mathbb{R}^{N \times N}$ and vector $\mathbf{c}_i, \mathbf{b}_i \in \mathbb{R}^{N \times 1}$ as time-varying parameters of selective SSMs. This formulation reveals that the mixer matrices $\mathbf{M}$ follow a fundamental class of structured matrices known as semiseparable matrices, defined as follows:

**Definition 3.1** (The rank characterization of semiseparable matrices). *A lower triangular matrix $\mathbf{M}$ is $N$-semiseparable iff any submatrix from the lower triangle (on or below the diagonal) has a rank of at most $N$. See Figure 2 (a).*

However, a key limitation of these matrices – and by extension, SSMs – is their inherent causality, which restricts their use in scenarios where bidirectional processing is vital. To circumvent this limitation, previous efforts [44, 15, 12] have explored employing two separate SSMs, one for forward and the other for backward sequence processing, then combine the outputs using strategies like element-wise addition, the Hadamard product, or concatenation. Among such heuristics, addition-based bidirectional extensions of SSMs [15, 48, 38] can be conceptualized within our matrix mixer framework, as illustrated in Figure 2 (c).

## 3.2 Quasiseparable Matrices: A Principled Bidirectional Matrix Mixer

We fully utilize the matrix mixer framework, which is discussed in Section 2, to explore a novel bidirectional sequence mixer and identify quasiseparable matrices as a prime candidate. Our exploration focuses on structured matrix classes that meet the following criteria: 1) they feature upper triangular components for bidirectionality, 2) they possess strong expressivity, and 3) they benefit from sub-quadratic matrix multiplication algorithms.

The structural design of quasiseparable matrices inherently meets the first criterion, which is defined as follows: a matrix $\mathbf{M}$ is $N$-quasiseparable if each element $m_{ij}$ satisfies

$$
m_{ij} = \begin{cases} \overrightarrow{\mathbf{c}_i^T} \overrightarrow{\mathbf{A}_{i:j}^{\times}} \overrightarrow{\mathbf{b}_j}, & \text{if } i > j \\ \delta_i, & \text{if } i = j \\ \overleftarrow{\mathbf{c}_i^T} \overleftarrow{\mathbf{A}_{i:j}^{\times}} \overleftarrow{\mathbf{b}_j}, & \text{if } i < j \end{cases}
\tag{3}
$$

where each $\delta_i$ is a scalar, $\mathbf{b}_i, \mathbf{c}_i \in \mathbb{R}^{N \times 1}$, and $\mathbf{A}_i \in \mathbb{R}^{N \times N}$ [4]. Clearly, this matrix class features non-zero upper triangular components, enabling bidirectional processing.

Furthermore, the second requirement – the expressivity of quasiseparable matrices – is confirmed by their rank characterization:

**Definition 3.2** (The rank characterization of quasiseparable matrices [30]). *A matrix* $\mathbf{M}$ *is* $N$*-quasiseparable iff any submatrix from either the strictly upper or lower triangle (off from the diagonal) has a rank of at most* $N$*. See Figure 2 (b).*

This definition emphasizes the rank constraint inherent in quasiseparable matrices, which is also evident from Equation (3) given that each vector $\mathbf{c}_i, \mathbf{b}_i \in \mathbb{R}^{N \times 1}$ and matrix $\mathbf{A}_i \in \mathbb{R}^{N \times N}$ has a rank of at most $N$. This structural flexibility of quasiseparable matrices directly leads to significant generalizations, extending the capabilities of both low-rank and semiseparable matrices.

**Corollary 3.3.** *Quasiseparable matrices generalize low-rank matrices.*

**Corollary 3.4.** *Quasiseparable matrices generalize and extend semiseparable matrices.*

Additionally, we revisit the previous addition-based bidirectional extensions of SSMs [15, 48, 38] through the lens of our matrix mixer framework. Unlike other elements, the diagonal values in a mixer matrix embody a unique concept of residuals, serving as a critical aspect of model expressivity. As demonstrated in Figure 2 (c), the mixer matrices in these bidirectional SSMs exhibit constraints in their diagonal elements $\{\overrightarrow{\mathbf{c}_i^T}\overrightarrow{\mathbf{b}_i} + \overleftarrow{\mathbf{c}_i^T}\overleftarrow{\mathbf{b}_i}\}_L$, which are directly influenced by the shared non-diagonal construction vectors $\{\overrightarrow{\mathbf{c}_i}, \overrightarrow{\mathbf{b}_i}, \overleftarrow{\mathbf{c}_i}, \overleftarrow{\mathbf{b}_i}\}_L$. Importantly, the rank characterization of semiseparable matrices includes *on-diagonal* elements, whereas that of quasiseparable matrices applies only to *off-diagonal* submatrices. This generosity in the rank-based definition suggests that sequence models employing quasiseparable mixers not only offer inherent extendability in handling both causal and bidirectional processing, but also exhibit strong expressivity.

**Corollary 3.5.** *Quasiseparable matrices are strictly more expressive than mixer matrices of addition-based bidirectional SSMs.*

Leveraging this inherent flexibility of quasiseparable matrix mixers, our Hydra in Section 3.3 is defined by incorporating shift operations. Our experimental results strikingly confirm that this nuanced parameterization difference leads to a notable improvement in downstream task performance, thereby substantiating our theoretical claims (see Appendix D.1).

Moreover, with their structural similarity to semiseparable matrices, quasiseparable matrices are confirmed as sequence aligned matrices. Given our experimental results that SAM parameterizations are the key to the strong representational power (Section 4.1.1), we further validate our choice of quasiseparable matrices for the bidirectional sequence mixer.

**Proposition 3.6.** $N$*-quasiseparable matrices are sequence aligned matrices.*

*Proof.* quasiseparable matrices, due to their structural similarity to semiseparable matrices, belong to the class of Sequence Aligned matrices. Specifically, the set of parameters is given by $\mathcal{P} = \hat{\mathcal{P}} = \{\mathbf{A}_i, \mathbf{b}_i, \mathbf{c}_i, \delta_i\}_L$. We consider the partition $\Pi = \{\{\mathbf{A}_i\}_L, \{\mathbf{b}_i\}_L, \{\mathbf{c}_i\}_L, \{\delta_i\}_L\}$, and for each element of the partition set, we choose the bijection that maps token $i$ to $\mathbf{A}_i$, $\mathbf{b}_i$, $\mathbf{c}_i$, and $\delta_i$ respectively. Finally, it is easy to see that the sub-matrix $M[:i+1,:i+1]$ indeed only contains parameters in the set $\{\mathbf{A}_j, \mathbf{b}_j, \mathbf{c}_j, \delta_j\}_i$, thus satisfying the last requirement of SAM matrices. $\qquad\square$

In Section 3.3, we detail how quasiseparable sequence mixer can be effectively implemented using existing SSM frameworks to achieve sub-quadratic multiplication efficiencies, fulfilling the final criterion of our matrix mixer exploration.

### 3.3 Taming the Hydra

As a direct consequence of the favorable mathematical properties of quasiseparable matrices, we present a new sequence mixer **Hydra**. We adopt quasiseparable matrices as matrix mixers in Hydra, which bring forth three significant advantages: 1) higher representational power compared to its heuristic alternatives, 2) easy to implement sub-quadratic matrix multiplication algorithm, and 3) significant parameter savings.

We exploit the relationship between semiseparable and quasiseparable matrices to develop an easy-to-implement, sub-quadratic matrix multiplication algorithm. Specifically, we recognize that quasiseparable matrices can be expressed as a combination of two semiseparable matrices.

**Proposition 3.7.** *Let* $\mathbf{X} \in \mathbb{R}^{L \times D}$ *be the input sequence, and let* $QS(\cdot)$ *and* $SS(\cdot)$ *denote the action of a quasiseparable and semiseparable matrix respectively. Let the two matrices share the parameters*

Table 2: **Matrix mixer ablations.** A systematic empirical study of matrix mixers on the GLUE benchmark by controlling for the architecture and varying only the matrix parameterization. Sequence-aligned matrices dynamically parameterize via input projections, becoming data-dependent (DD) that significantly increases performance. Most DD variants achieve competitive GLUE scores.

| Structure | DD | #Params | C4 Pretrain | | GLUE Tasks | | | | | | | | GLUE |
|---|---|---|---|---|---|---|---|---|---|---|---|---|---|
| | | | $\mathcal{L}_{ce}$ | Acc | MNLI | QNLI | QQP | RTE | SST2 | MRPC | COLA | STS | Avg |
| Dense | | 71M | 2.05 | 59.6 | 73.3 | 76.2 | 85.3 | 64.4 | 90.8 | 84.7 | 45.7 | 76.8 | 74.7 |
| Toeplitz | | 71M | 1.97 | 60.8 | 74.6 | 79.6 | 86.6 | 66.1 | 90.9 | 84.2 | 45.7 | 79.1 | 75.8 |
| Toeplitz | ✓ | 72M | 1.91 | 61.9 | 77.3 | 81.8 | 88.1 | 67.1 | 90.7 | 87.3 | 45.3 | 84.1 | 77.7 |
| DFT | | 71M | 2.46 | 53.1 | 70.4 | 70.8 | 84.5 | 59.9 | 89.8 | 83.6 | 44.4 | 69.8 | 71.7 |
| Vandermonde | | 71M | 2.46 | 53.0 | 55.2 | 61.3 | 82.5 | 66.3 | 87.4 | 84.2 | 45.8 | 84.2 | 70.8 |
| Vandermonde | ✓ | 70M | 2.04 | 59.7 | 74.1 | 80.0 | 86.2 | 67.9 | 89.3 | 84.3 | 46.0 | 80.1 | 76.0 |
| Cauchy | | 71M | 2.25 | 56.2 | 75.3 | 81.3 | 86.7 | 66.6 | 88.8 | 84.5 | 27.4 | 83.2 | 74.2 |
| Cauchy | ✓ | 70M | 1.94 | 61.6 | 77.5 | 84.4 | 84.2 | 68.0 | 91.0 | 86.7 | 48.1 | 85.2 | 78.2 |
| Low-Rank | | 71M | 2.06 | 59.4 | 73.7 | 76.5 | 85.1 | 61.5 | 90.6 | 85.8 | 49.2 | 76.6 | 74.9 |
| Low-Rank | ✓ | 70M | 1.90 | 62.2 | 77.6 | 84.1 | 88.2 | 69.1 | 91.0 | 85.9 | 47.6 | 83.9 | 78.4 |
| Attention | | 71M | 2.08 | 59.1 | 71.0 | 70.4 | 83.5 | 62.3 | 89.9 | 83.3 | **49.6** | 65.2 | 71.9 |
| Attention | ✓ | 70M | 1.87 | 62.9 | 78.5 | 85.4 | 88.4 | 67.9 | 91.2 | 86.4 | 47.8 | 84.3 | 78.8 |
| Quasiseparable | | 72M | 2.03 | 59.8 | 73.8 | 78.1 | 87.1 | 64.3 | 90.2 | 84.4 | 45.5 | 77.2 | 75.1 |
| Quasiseparable | ✓ | 71M | **1.84** | **63.3** | **79.5** | **85.5** | **88.6** | **69.8** | **91.9** | **88.4** | 48.4 | **85.6** | **79.7** |

$\{\mathbf{A}_i, \mathbf{b}_i, \mathbf{c}_i\}_L$, *and define* $\mathbf{D} = diag(\delta_1, \cdots, \delta_L)$, *where* $\delta_i$*'s are the diagonal parameters of the quasiseparable matrix. Then,*

$$QS(\mathbf{X}) = shift(SS(\mathbf{X})) + flip(shift(SS(flip(\mathbf{X})))) + \mathbf{D}\mathbf{X},$$

*where flip(·) denotes the operation that reverses the input sequence, while shift(·) refers to shifting the sequence right by one position, padding the beginning with zero. (Proof in Appendix B)*

The above proposition demonstrates that quasiseparable matrix multiplication can be decomposed into two operations of semiseparable matrix multiplication with simple functions such as `flip` and `shift`. Given that semiseparable matrix structure encompasses SSMs, this flexibility allows for the selection of any SSM variant for implementation. In this paper, we employ SSD [6], chosen for its linear-time and dedicated hardware-efficient implementation. However, the architecture of Hydra is compatible with a variety of SSMs [18, 17, 16, 6] and can also be generalized with other recurrent models [47, 28].

Furthermore, Hydra significantly improves parameter over the heuristic approaches to bidirectional modeling using SSMs [15, 48, 44, 12]. For example, some approaches utilize two distinct SSMs, which doubles the number of training parameters. In contrast, since we conceptualize the model as a quasiseparable matrix mixer (see Figure 4), we naturally share the $f_X$ projection layer, which accounts for a bulk of the model size. Empirically, we observe only a marginal increase in the total number of parameters compared to a single SSM, and can cut the the number of parameters nearly in half compared to the heuristic approaches to bidirectionality.

## 4 Experiments

In Section 4.1, we begin by analyzing the matrix mixer framework through extensive performance comparisons between different structured matrix classes (Section 4.1.1). The data-dependent parameterization of Quasiseparable matrices surpasses the performance of all other matrix classes, validating our selection of it as the sequence mixer. Furthermore, we compare our method against other ad-hoc solutions that extend SSMs to acquire bidirectionality (Appendix D.1), and show that our Hydra outperforms them, underscoring the utility of the matrix view of sequence mixers.

Then, in Section 4.2, we validate the effectiveness of our method by evaluating Hydra on quintessential language and image benchmark tasks: Masked Language Modeling (Section 4.2.1) and Image Classification (Section 4.2.2). State-of-the-art performance in both tasks has generally been dominated by transformer-based models [10, 11]. Our Hydra serves as an efficient and powerful replacement to the transformer layer, outperforming it on both tasks.

In the presentation of results across all tables, the highest performing scores are highlighted in **bold**, while the second-highest scores are marked with an underline. Each number is the average of five runs.

Table 3: Performance of various approaches that extend Mamba to a bidirectional model. We compare our quasiseparable matrix mixer to element-wise addition (Add), the Hadamard product (Mult), and concatenation (Concat) variants.

| Method | #Params | C4 Pretrain | | GLUE |
| | | $\mathcal{L}_{ce}$ | Acc (%) | Avg |
| --- | --- | --- | --- | --- |
| Mamba | 68M | 3.45 | 39.5 | 77.7 |
| Add | 70M | 1.68 | 65.6 | 80.6 |
| Mult | 70M | 1.72 | 64.9 | 79.6 |
| Concat | 69M | 1.71 | 65.4 | 81.1 |
| Quasi | 70M | **1.66** | **65.9** | **81.7** |

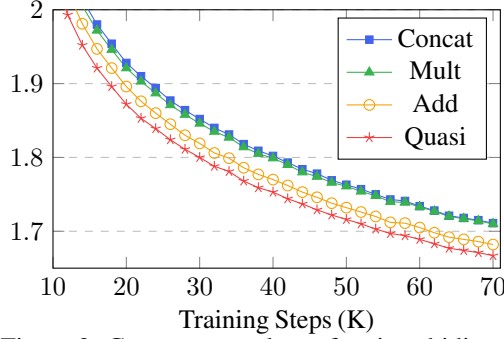

Figure 3: Cross-entropy loss of various bidirectional variants, measured on the C4 validation set.

## 4.1 Analysis of the Matrix Mixer Framework

### 4.1.1 Effects of Different Structured Matrix Families

Our controlled experimental setting distinctly separates the mixer matrices from other architectural components, enabling a rigorous and focused comparison between different types of mixer matrices. Specifically, utilizing the recent Mamba [6] block, we only replace SSD with different mixer matrices M. In Table 2 we provide experimental results that showcase the expressivity of various matrix mixers that support bidirectional sequence processing, primarily by utilizing off-the-shelf structured matrix families for the mixer matrix. Further details of the experimental setup are provided in Appendix E.

**Results.** The results distinctly highlight the substantial advantage in expressivity conferred by the SAM property (Definition 2.2). Previously, the high performance observed in [22, 45] was attributed to their low-rank based mixer matrices, particularly emphasized by the query-key $QK^T$ interaction. The results show that the structures of mixer matrices affect the capability of sequence models. However, we demonstrate that the key factor that primarily contributes to significant improvements in the expressivity of sequence models is not the query-key formulation, but rather the SAM property. Through our systematic extension, we adapt six structured matrix families – Toeplitz, Vandermonde, Cauchy, low-rank, Attention, and quasiseparable – to include the SAM property (see Appendix E for implementation details). This adaptation reveals that the sequence-aligned Cauchy variant performs nearly as well as the sequence-aligned low-rank variant, which characterizes LA. Moreover, the experimental results consistently indicate that variants equipped with the SAM property outperform those lacking this feature across all tested matrix families.

### 4.1.2 Ablating Approaches for Bidirectionality

We compare the quasiseparable matrix mixer approach to prior bidirectional SSMs models that achieve bidirectionality by aggregating forward and backward SSMs using various heuristics including element-wise addition [15], the Hadamard product [44], and concatenation [44, 12]. We provide details of the ablation studies in Appendix D.1.

**Results.** The results presented in Table 3 show the shortcomings of the unidirectional Mamba [16] when applied to bidirectional contexts, as evidenced in C4 and GLUE. This significant performance disparity $(-4.0)$ underscores the essential need for models to be capable of bidirectional sequence processing. Within the comparison of four bidirectional variants shown in Table 3, our approach of utilizing a quasiseparable matrix achieves the top validation results on the C4 benchmark and records the highest GLUE average score of $81.7$. This advantage of our method is further validated in Figure 3, where it demonstrates the cross-entropy loss on the C4 validation set throughout training. Considering the gigantic size of the dataset, the expressivity of Hydra using quasiseparable matrices is clearly manifested by consistently achieving the lowest loss, as well as the highest masked token prediction accuracy.

## 4.2 Evaluation Results of Hydra

### 4.2.1 Bidirectional Masked Language Modeling

We pretrain our models on the masked language modeling objective using the Colossal Cleaned Common Crawl (C4) corpus [36], then finetune and evaluate them on the GLUE benchmark [43]. We relegate experimental details in Appendix D.2.

Table 4: **GLUE Results.** Evaluation of various sequence models that can be formulated as matrix mixers. For maximum performance, all models are trained using established recipes [32, 13].

| Method | #Params | C4 Pretrain | | GLUE Tasks | | | | | | | | GLUE Avg |
|---|---|---|---|---|---|---|---|---|---|---|---|---|
| | | $\mathcal{L}_{ce}$ | Acc (%) | MNLI | QNLI | QQP | RTE | SST2 | MRPC | COLA | STS | |
| BERT | 110M | 1.59 | 67.3 | 84.4 | **90.3** | 89.7 | 77.1 | 92.3 | 90.7 | 54.2 | **89.1** | 83.5 |
| MLP-Mixer | 112M | 1.77 | 63.5 | 77.2 | 82.4 | 87.6 | 67.3 | 90.5 | 86.5 | 43.0 | 85.2 | 77.5 |
| FNet | 112M | 1.94 | 61.3 | 74.9 | 82.1 | 85.7 | 63.6 | 87.6 | 86.4 | 42.7[2] | 83.1 | 75.8 |
| M2 | 116M | 1.65 | 65.9 | 80.5 | 86.0 | 87.0 | 69.3 | 92.3 | 89.2 | 56.0 | 86.9 | 80.9 |
| Hydra | 112M | **1.46** | **69.1** | **84.5** | 90.0 | **91.3** | **77.5** | **93.5** | **91.2** | **57.2** | 88.9 | **84.3** |

**Results.** The results in Table 4, showcase that Hydra outperforms all existing state-of-the-art methods. Notably, Hydra surpasses the performance of BERT – trained with the latest HuggingFace recipe [46] – in both pretraining and GLUE benchmark scores. BERT has shown a noticeable gap in the masked language modeling task compared to other previous methods [40, 23, 13]. Hydra gains a 1.8% improvement in accuracy of C4 validation and a 0.8% increase in the average GLUE score, illustrating the effectiveness of leveraging matrix mixer view for bidirectional settings.

#### 4.2.2 Image Classification

We assess Hydra on the renowned ImageNet-1K benchmark [9], which comprises 1.28M training images and 50k validation images across $1,000$ categories. We use the ViT-Base [11] model as a baseline to facilitate a rigorous comparison of various sequence mixers by substituting the Transformer block in ViT with alternative sequence mixer models, specifically S4ND [27], Hyena [31], Mamba [16], and our proposed Hydra model. Unlike many off-the-shelf models such as CNN-based [19, 25] and vision-specialized Transformers [24] that include additional

Table 5: Top 1 & 5 image classification accuracies evaluated on the ImageNet-1K benchmark. We also report accuracies using the common model ensembling technique: Exponential Moving Average (EMA) weights. (Top) Reported from literature [27, 31]. (Bottom): Our unidirectional and bidirectional Mamba results.

| Method | #Params | Top-1 (%) | | Top-5 (%) | |
|---|---|---|---|---|---|
| | | Acc | Acc$_{EMA}$ | Acc | Acc$_{EMA}$ |
| ViT-B | 87M | 78.8 | 80.6 | 94.2 | 95.2 |
| S4-ViT-B | 89M | 79.4 | 80.4 | 94.2 | 95.1 |
| Hyena-ViT-B | 88M | 78.4 | 76.4 | 94.0 | 93.0 |
| Mamba-ViT-B | 89M | 79.1 | 80.0 | 94.2 | 94.9 |
| Hydra-ViT-B | 91M | **81.0** | **81.6** | **95.3** | **95.6** |

techniques such as hierarchical spatial downsampling to boost accuracy, our approach involves substituting only the sequence mixer layers within the ViT architecture. In addition, as opposed to other baselines in the setting of [31], our method uses **no tuning over the default ViT recipe** except for droppath. We found that Hydra fits the training data noticeably better than ViT, perhaps due to better expressivity and inductive bias, so we simply increased droppath from 0.3 to 0.5 as stronger regularization. We relegate further experimental details in Appendix D.2.

**Results.** The results, as presented in Table 5, compare the performance of Hydra with ViT [11] and other variants [27, 31] on ImageNet-1K. Hydra exhibits superior performance in image classification, outperforming ViT by 2.2% in Top-1 accuracy and 1.1% in Top-5 accuracy. Remarkably, even though Hydra simply flattens images without incorporating any specific 2D architectural adjustments, it still surpasses S4ND [27] – which is specifically tailored for image processing – by a notable margin of 1.6% in Top-1 accuracy. This showcases the versatility and effectiveness of Hydra in handling diverse data types.

## 5   Conclusion

In this work, we have explored a common paradigm for sequence models wherein the sequence mixer can be represented by a matrix. This framework encompasses many well-known models such as MLP-Mixer, FNet, convolutions, Transformers (softmax attention), and recent state-space models such as Mamba. By formalizing the matrix mixer framework and exploring additional matrix variants, we have identified a key axis of variation (*sequence alignment*) in matrix parameterizations, which enables benefits such as data dependence. This, in turn, provides increased flexibility and stronger performance for sequence models. Furthermore, we have leveraged the matrix mixer framework to motivate a natural bidirectional extension of state space models called Hydra, which can be formulated as *quasiseparable* matrix mixers. Hydra consistently outperforms unidirectional Mamba and other bidirectional sequence models in tasks such as masked language modeling and image classification.

---

[2]We adjust the learning rate to $1e-5$ to address instabilities observed in the training of FNet on COLA.

## Acknowledgements

This research was made possible by the generous support of computational resources provided by Cartesia AI.

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

# A  Discussions

## A.1  Limitations

In this section, we share two limitations of our work, namely 1) Representation-Computation Tradeoff, and 2) Hardware Efficiency.

**Representation-Computation Tradeoff.**    While structured matrix mixers are computationally more efficient than their dense matrix mixer counterparts like softmax attention, they are also representationally less expressive, which may be seen as a limitation of these methods. For instance, concurrent works [1, 2, 21] have begun investigating the representational power of SSMs by analyzing their performance on memorization-centric tasks. They report that SSMs with a fixed model capacity are eventually outperformed by softmax attention for longer sequences. This can be viewed as a consequence of the matrix being too structured, and hence too inexpressive for the problem.

On the other hand, we remark that the degree of structure of a structured matrix is a knob that can be tuned according to the specific task, that is we can tradeoff the computational efficiency of the method for larger expressivity. For instance, within the structured matrix class of low rank matrices, we can tune the rank upto the size of the matrix, which is the sequence length. As the rank of the matrix class increases so does its expressive power; however, at the same time it also diminishes the compute efficiency associated with the matrix being low rank.

As another example, we demonstrate this tradeoff on the performance of SSMs on retrieval-style tasks for SSD, which is the modern variant of Mamba. Specifically, we show that SSD is able to recover the accuracy attained by softmax attention once we control for the compute capacity of the model. In contrast, hardware limitations of the selective scan algorithm make it impractical to match the compute capacity in Mamba, explaining the emerging findings from [1, 2, 21] that SSMs underperform on memorization-centric tasks. This makes it evident that the development and analysis of SSMs is an active area of research with substantial room for exploration and improvement.

**Hardware Efficiency:**    Despite the fact that structured matrices have associated sub-quadratic matrix multiplication algorithms, their implementation may not be hardware-friendly, which can reduce the execution speed in practice. For instance, one of the core advantages of Transformer is the hardware-friendly nature of its architectural design, mainly composed of pure matrix multiplications.

## A.2  Broader Impact

This paper presents work whose goal is to advance the field of Machine Learning. The primary objective of this work is to introduce novel component designs that are universally applicable across various tasks. Consequently, we believe that this work is unlikely to have significant societal impacts that necessitate special emphasis within this context.

# B  Proofs

**Proposition 2.5.** *MLP-Mixer is a Matrix Mixer*

*Proof.*  Recall that MLP-Mixer applies a weight-shared, per-channel linear projection on the input sequence. Since MLP-Mixer does not pre-process the input sequence, we set $f_X$ to be the identity function. Additionally, since the mixer matrix is data independent and is learned as a free parameter, we define the class $\mathcal{M} := \mathbb{R}^{L \times L}$, and set $f_{\mathcal{M}}(\mathbf{X};\theta) := \theta^3$, for $\theta \in \Theta$, where $\Theta := \mathbb{R}^{L \times L}$. Since MLP-Mixer shares the matrix across all channels, the number of heads $H = 1$. The output is given by $\mathbf{Y}^{(0)} = \theta \mathbf{X}^{(0)}$, where $\theta$ is the learned mixer matrix, which matches with the functional form of MLP-Mixer's sequence mixing primitive.  □

**Proposition 2.6.** *LA is a Structured Matrix Mixer*

*Proof.*  Recall that for each head $h \in [H]$, LA computes,
$$\mathbf{Y}^h = \sigma(\mathbf{Q}^h)\sigma(\mathbf{K}^h)^T \mathbf{V}^h,$$
where $\mathbf{Q}^h, \mathbf{K}^h, \mathbf{V}^h$ and projections of the input. Since LA preprocesses the input sequence via projection, we set $f_X = \mathbf{W}_V \in \mathbb{R}^{C \times D}$, with $\mathbf{W}_V$ as a learned parameter. The class of structured matrices $\mathcal{M}$

---

[3]For simplicity, we overload the notation to denote a finite-dimensional linear map with its corresponding matrix.

is clearly the class of Low Rank matrices. We define the set $\Theta = (\mathbb{R}^{C \times Hd})^2$ to correspond to the learned parameters $\mathbf{W}_Q, \mathbf{W}_K$ for the $\{\mathbf{Q}^h, \mathbf{K}^h\}_h$ matrices, where $d$ is the internal dimension for keys and queries. We set the matrix-generating function $f_\mathcal{M}^h$ to compute $\mathbf{M}^h$ as $\mathbf{M}^h = \sigma(\mathbf{Q}^h)\sigma(\mathbf{K}^h)^T$ through linear projections, followed by applying element-wise nonlinearity $\sigma$, and a matrix multiplication. The output $\mathbf{Y}^h = \mathbf{M}^h(f_X(\mathbf{X}))^h$ matches the expected functional form of LA's sequence mixing primitive. $\square$

**Proposition 3.2.** *Let $\mathbf{X} \in \mathbb{R}^{L \times D}$ be the input sequence, and let $QS(\cdot)$ and $SS(\cdot)$ denote the action of a Quasiseparable and Semiseparable matrix respectively. Let the two matrices share the parameters $\{\mathbf{A}_i, \mathbf{b}_i, \mathbf{c}_i\}_L$, and define $\mathbf{D} = diag(\delta_1, \cdots, \delta_L)$, where $\delta_i$'s are the diagonal parameters of the Quasiseparable matrix. Then,*
$$QS(\mathbf{X}) = shift(SS(\mathbf{X})) + flip(shift(SS(flip(\mathbf{X})))) + \mathbf{DX},$$
*where $flip(\cdot)$ denotes the operation that reverses the input sequence, while $shift(\cdot)$ refers to shifting the sequence right by one position, padding the beginning with zero.*

*Proof.* The proof follows by observing that the first term, $shift(SS(\mathbf{X}))$, models the effect of the lower triangular part of the QS matrix. The next term, $flip(shift(SS(flip(\mathbf{X}))))$, represents the upper triangular part of the QS matrix, and the last term, $\mathbf{DX}$, accounts for the QS matrix's diagonal. $\square$

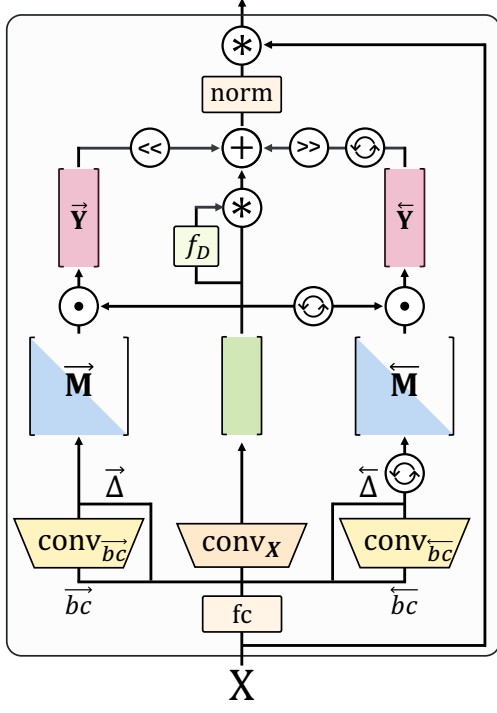

Figure 4: Detailed illustration of Hydra.

```
def hydra(
    x,          # (B,L,H*P)
    A           # (H,) Parameter
):
    x_b = flip(x, dim=1)

    # (B,L,H)
    dt_f, dt_b = proj_dt(x), proj_dt(x_b)

    # (B,L,H*P)
    y_f = SSD(
        x,
        discretize_A(A, dt_f),      # (B,L,H)
        discretize_bc(x, dt_f),     # (B,L,N)
    )
    y_b = SSD(
        x_b,
        discretize_A(A, dt_b),      # (B,L,H)
        discretize_bc(x_b, dt_b),   # (B,L,N)
    )

    y_f = shift(y_f, dim=1)
    y_b = flip(shift(y_b, dim=1), dim=1)

    y = y_f + y_b + x * repeat(
        proj_D(x),
        "B L H -> B L (H P)"
    )
    return y
```

Figure 5: Pseudo code for Hydra. $B, L, H, P$ denote batch size, sequence length, number of heads, and head dimension respectively. The suffices `_f` and `_b` denote **f**orward and **b**ackward. `shift`: Right-shift.

## C  Architectural Details

Here, we provide architectural specifics of our method Hydra. As shown in Figure 4 and previously discussed Figure 5, Hydra is largely divided into two sub-components, each for forward and backward sequence processing respectively. A key architectural difference from the unidirectional Mamba [16] and SSD [6] models is the type of convolutional layers employed. Unlike Mamba and SSD, which utilize causal convolutions due to their unidirectional nature, Hydra incorporates standard convolutional layers, reflecting its capacity for bidirectional sequence processing.

## D  Additional Details

### D.1  Details of Ablation Studies in Section 4.1

**Bidirectionality.** We provide details of the experiments in . Each method under consideration employs 12 layers. As depicted in Table 3, due to shared projection layers for forward and backward

SSMs, Hydra only requires an additional 2M parameters compared to its unidirectional counterpart. Moreover because of the parameter increase in the concatenation variant, we lower its hidden dimension to match parameters. As `[CLS]` token is typically placed at the start of a sequence, we add an additional technique only to Mamba, which substitutes a `[CLS]` token with a global average pool of tokens.

## D.2 Details of Main Results in Section 4.2

**Training Details of Hydra** We align our setup with the BERT-Base [10] architecture, which consists of 110M parameters across 12 transformer encoder layers. To ensure a fair comparison, MLP-Mixer [40], FNet [23], and M2 [13] are configured with 12 layers, with the number of channels adjusted to match the parameter count of BERT. Similarly, Hydra is structured with 23 layers to align with the total number of parameters of BERT. We follow the well-established BERT training recipes [32, 13] for optimal performance of each method. The specific hyperparameters for reproducing the results in Table 4 are reported in Table 6, and the settings used for obtaining the results of Hydra in Table 5 are listed in Table 9.

**GLUE.** We pretrain on the C4 dataset, and follow the recipe from MosaicBERT [32] which has been widely adopted in recent works - the models are trained for $70k$ steps with a batch size of $4096$. We adopt the same hyperparameters as M2 [13] for pretraining, including the optimizer, learning rate schedule, and the bert-base uncased tokenizer. We employ the controlled setting from [20] for finetuning models on the GLUE benchmark. Specifically, we utilize weights pretrained on the C4 corpus for the five tasks – MNLI, QNLI, QQP, SST2, and COLA – and finetune RTE, STSB, and MRPC from an MNLI checkpoint. For reliability, the performance metrics reported represent the average scores from five runs for each GLUE task, with seeds selected completely at random.

The well-established BERT and M2 models benefit from highly optimized training and finetuning recipes [20, 13]. Therefore, we perform a short sweep for the learning rate and number of epochs for finetuning tasks to obtain the best results. To note, we ensure that the number of epochs do not surpass their original values for fairness. In Table 7 first row, we observe that Hydra of $24$ layers trained using the M2 recipe out-of-the-box outperforms both BERT ($83.5$) and M2 ($80.9$) on the GLUE benchmark without any hyperparameter tuning. From the tuned hyperparameter, Hydra with 23 layers (second row) gains additional improvements, outperforming the results obtained by using the M2 recipe.

In Table 8, we report extended results for our GLUE comparisons (Table 4), including reference numbers directly reported from the literature. We include these to show that our results are a fair comparison of different models using our internally-consistent training framework, and that our baselines are strong and generally consistent with those found in MosaicBERT [32].

**ImageNet-1K.** For Table 5, we use $12$ layers for the Transformer [42], S4ND [27], and Hyena [31] blocks, and set to $24$ layers for Mamba and Hydra to match the number of parameters. The training involves images of $224 \times 224$ resolution, tokenized using a $16 \times 16$ patchifying layer. For Hydra, we adopt a row-major ordering to flatten patches and deliberately exclude positional embeddings. The models are trained over 300 epochs, leveraging hyperparameters primarily sourced from a standard training recipe [24], including an initial learning rate of $1e-4$ and a batch size of $1024$. We also incorporate regular ImageNet-1K training techniques [41], such as augmentation and regularization strategies. All other configurations remain consistent with the original ViT [11] setup, ensuring that any observed differences in performance can be directly attributed to the expressivity of the respective sequence mixer layers.

## E Additional Details of Ablation Studies in Table 2

To rigorously demonstrate the representational power of various families of matrix mixers, we ensure a fair comparison by meticulously controlling architectural hyperparameters. Specifically, all variants are constructed on the SSD [6] block, with a consistent configuration of 12 layers, an expansion factor of 2, and a hidden dimension (`d_model`) of 768. We then adjust the channel dimensions for `qk_dim` and `headdim` to keep the total parameter counts close to 70M. The models are pretrained for $24k$ steps in C4, each with a 4096 batch size, thus processing approximately 12.5B tokens in total. The finetuning phase adheres to the standardized protocol used for Hydra, ensuring comparability.

The primary differentiating factor among the variants is the mixer matrix $\mathbf{M}$, which has two main configurable attributes: 1) the presence of the SAM property, and 2) the family of a mixer matrix. According to Definition 2.2, mixers with the SAM attribute contain parameters that are dynamically generated from elements of the input sequence, making them input data-dependent (Proposition 2.3). In contrast, variants without SAM are data-independent, with shared parameters across all inputs. The

Table 6: Hyperparameters of different recipes used for training GLUE benchmark tasks.

| | | MNLI | QNLI | QQP | RTE | SST2 | MRPC | COLA | STS |
|---|---|---|---|---|---|---|---|---|---|
| [20] | LR | 5e-5 | 1e-5 | 3e-5 | 1e-5 | 3e-5 | 8e-5 | 5e-5 | 3e-5 |
| | WD | 5e-6 | 1e-6 | 3e-6 | 1e-6 | 3e-6 | 8e-6 | 5e-6 | 3e-6 |
| | n_epochs | 3 | 10 | 5 | 3 | 3 | 10 | 10 | 10 |
| | seq_len | 256 | 256 | 256 | 256 | 256 | 256 | 256 | 256 |
| [13] | LR | 5e-5 | 5e-5 | 3e-5 | 1e-5 | 3e-5 | 5e-5 | 5e-5 | 7e-5 |
| | WD | 5e-6 | 1e-6 | 1e-2 | 1e-2 | 3e-6 | 1e-2 | 5e-6 | 1e-2 |
| | n_epochs | 3 | 10 | 10 | 6 | 3 | 10 | 10 | 10 |
| | seq_len | 128 | 128 | 128 | 128 | 128 | 128 | 128 | 128 |
| Ours | LR | 1e-4 | 5e-5 | 5e-5 | 1e-5 | 5e-5 | 8e-5 | 1e-4 | 3e-5 |
| | WD | 5e-6 | 1e-6 | 3e-6 | 1e-6 | 3e-6 | 8e-6 | 5e-6 | 3e-6 |
| | n_epochs | 2 | 7 | 3 | 3 | 2 | 10 | 10 | 10 |
| | seq_len | 256 | 256 | 256 | 256 | 256 | 256 | 256 | 256 |

Table 7: Evaluation of Hydra on C4 dataset and GLUE Benchmark using different training recipes. Specific hyperparameters for reproducing the results are provided in Table 6.

| Recipe | #Params | Pretrain | | GLUE Tasks | | | | | | | | GLUE |
|---|---|---|---|---|---|---|---|---|---|---|---|---|
| | | $\mathcal{L}_{ce}$ | Acc (%) | MNLI | QNLI | QQP | RTE | SST2 | MRPC | COLA | STS | Avg |
| [13] | 115M | 1.45 | 69.3 | 83.7 | 89.7 | 89.7 | 77.4 | 92.8 | 91.5 | 54.7 | 90.1 | 83.7 |
| Ours | 112M | 1.46 | 69.1 | 84.5 | 90.0 | 91.3 | 77.5 | 93.5 | 91.2 | 57.2 | 88.9 | 84.3 |

Table 8: Official GLUE benchmark results from the referenced papers [32, 13, 23].

| Method | Source | #Params | GLUE Tasks | | | | | | | | GLUE |
|---|---|---|---|---|---|---|---|---|---|---|---|
| | | | MNLI | QNLI | QQP | RTE | SST2 | MRPC | COLA | STS | Avg |
| BERT | Ours | 110M | 84.4 | 90.3 | 89.7 | 77.1 | 92.3 | 90.7 | 54.2 | 89.1 | 83.5 |
| | [32] | 110M | 84.1 | 89.8 | 91.2 | 77.2 | 91.2 | 87.5 | 54.6 | 88.9 | 83.2 |
| | [23] | 110M | 82.5 | 91.0 | 87.0 | 69.0 | 93.0 | 83.0 | 73.0 | 89.0 | 83.3 |
| | [13] | 110M | - | - | - | - | - | - | - | - | 79.6 |
| MLP-Mixer | Ours | 112M | 77.2 | 82.4 | 87.6 | 67.3 | 90.5 | 86.5 | 43.0 | 85.2 | 77.5 |
| FNet | Ours | 112M | 74.9 | 82.1 | 85.7 | 63.6 | 87.6 | 86.4 | 42.7 | 83.1 | 75.8 |
| | [23] | 83M | 72.5 | 80.0 | 83.0 | 63.0 | 95.0 | 76.0 | 69.0 | 79.0 | 76.7 |
| | [23] | 238M | 77.0 | 85.0 | 85.0 | 69.0 | 94.0 | 88.0 | 78.0 | 84.0 | 81.9 |
| M2 | [13] | 116M | 80.5 | 86.0 | 87.0 | 69.3 | 92.3 | 89.2 | 56.0 | 86.9 | 80.9 |
| Hydra | Ours | 112M | 84.5 | 90.0 | 91.3 | 77.5 | 93.5 | 91.2 | 57.2 | 88.9 | 84.3 |

inclusion of projection layers for data dependency results in a marginal increase in parameter count for models with SAM. Consequently, we precisely configure qk_dim $= 16$, headdim $= 128$ for SAM variants, and qk_dim $= 64$, headdim $= 64$ for non-SAM variants. For dense, Toeplitz (SAM and non-SAM), and Vandermonde DFT, we slightly adjust the hyperparameters to match the number of parameters.

To facilitate understanding and reproducability of these methods, we provide PyTorch codes in Figure 6, Figure 7, Figure 8, Figure 9, Figure 10, and Figure 11. Our primary focus on this ablation is the comparison of expressivity between different Matrix Mixers. Therefore, our implementations do not necessarily adopt algorithms with efficient computational complexities for simplicity. The abbreviation di stands for **d**ata-**i**ndependent and dd represents **d**ata-**d**ependent, in which dd variants are equipped with the SAM property. As the Quasiseparable variant is equivalent to the Hydra blocks, its implementation can be found in the Python file provided in the supplementary. Further specific details are described below.

**Dense.** While extending Dense matrices to incorporate the SAM attribute is not straightforward, implementing the vanilla Dense mixers (*i.e.* without SAM) is extremely simple: they are equivalent to MLP-Mixer [40], employing a mixer matrix of $\mathbb{R}^{L \times L}$.

Table 9: ViT settings for ImageNet-1K.

| Parameter | Value |
|---|---|
| Image size | $224^2$ |
| Optimizer | AdamW |
| Optimizer momentum | $\beta_1,\beta_2=0.9,0.999$ |
| Weight init | trunc. normal (std=0.02) |
| ViT base learning rate | $1e-3$ |
| ViT weight decay | 0.05 |
| Dropout | None |
| Batch size | 1024 |
| Training epochs | 300 |
| Learning rate schedule | cosine decay |
| Warmup epochs | 10 |
| Warmup schedule | linear |
| Randaugment | (9,0.5,layers=2) |
| Mixup | 0.8 |
| Cutmix | 1.0 |
| Random erasing | 0.25 |
| Label smoothing | 0.1 |
| Stochastic depth | 0.5 |
| Exp. mov. avg (EMA) | None |

**Toeplitz.** Toeplitz matrix mixers without SAM properties are equivalent to convolution operations [18, 17], formulated with $\mathbb{R}^{2L-1}$ parameters. We also present a Toeplitz matrix mixer with SAM properties, thus data-dependent and can handle sequences of arbitrary lengths. The core idea is fairly similar to the quasiseparable matrix mixer – Hydra – that the Toeplitz matrix mixer integrates outputs from two separate forward and reverse sequence convolutions. Specifically, for $i \in 0,...L-1$, each token $x_i$ generates two convolution parameters $m_{-i}$ and $m_i$. Using all $m$ parameters, a data-dependent dynamic convolutional weight is generated as $\{m_{-L+1},m_{-L+2},...,m_{-1},m_0,m_1,...m_{L-2},m_{L-1}\}$.

**Vandermonde.** In Section 2.4, we introduced the single-headed sequence aligned Vandermonde matrix mixer. Leveraging the definition, we present multi-headed implementation by a seamless extension.

**Cauchy.** The constant for preventing the denominator approaching zero is initialized to $0.0$ and $0.5$ for the di and the dd variants, respectively.

**Definition E.1** (Cauchy Matrix). *Given* $\mathbf{q},\mathbf{k} \in \mathbb{R}^L$, *a matrix* $\mathbf{M}$ *is Cauchy if each* $(i,j)$-*entry* $m_{ij}$ *satisfies* $m_{ij} = \frac{1}{q_i-k_j}$; $\quad q_i - k_j \neq 0$.

# F  Ablating the impact of Shift and Diagonal operations in Hydra

Observe that general Quasiseparable (QS) matrices can be viewed as the Add variant augmented by two operations: an extra diagonal component and a shift operation. In this ablation we assesses the impact of each operation on the model performance by individually removing either the diagonal or shift term, as well as removing both (Add) to analyze why QS outperforms Add.

We find that using only the shift (80.7) or only the diagonal (80.7) yields the same performance as the Add method (80.6). It is only when both operations are combined that we observe a clear improvement. This result validates our choice of QS matrices, as neither Add+Diag nor Add+Shift alone has the expressivity of QS which is strictly more expressive than the addition of two SS matrices.

Table 10: Analyzing the impact of Shift and Diagonal operations in Hydra.

| Method | #Params | Acc (%) | GLUE |
|---|---|---|---|
| Add | 70M | 1.68 | 80.6 |
| Add+Diag | 70M | 1.68 | 80.7 |
| Add+Shift | 70M | 1.67 | 80.7 |
| Quasi | 70M | 1.66 | 81.7 |

```python
class Dense(nn.Module):
    def __init__(
        self,
        d_model,
        max_seq_len, # max_seq_len is necessary for Dense.
        expand=2,
        headdim=128,
        device=None,
        dtype=None,
    ):
        factory_kwargs = {"device": device, "dtype": dtype}
        super().__init__()
        self.d_model = d_model
        self.max_seq_len = max_seq_len
        self.expand = expand
        self.d_inner = self.expand * self.d_model
        self.headdim = headdim
        assert self.d_inner % self.headdim == 0
        self.nheads = self.d_inner // headdim

        self.std_dev = 1 / np.sqrt(self.max_seq_len)

        self.M = nn.Parameter(
            torch.empty(self.nheads, self.max_seq_len, self.max_seq_len, **factory_kwargs
                        )
        )
        nn.init.xavier_normal_(self.M)

    def forward(self, hidden_states):
        residual = hidden_states
        # Rearrange hidden states to shape [batch, n_heads, length, headdim]
        hidden_states = rearrange(hidden_states, 'b l (n h) -> b n l h', n=self.nheads)

        output = torch.einsum('b n t h, n l t -> b n l h', hidden_states, self.M)
        output = self.std_dev * output
        output = rearrange(output, 'b n l h -> b l (n h)') + residual

        return output
```

Figure 6: PyTorch code for the Dense variant in Table 2.

```python
class Toeplitz(nn.Module):
    def __init__(
        self,
        is_data_dependent,
        d_model,
        max_seq_len, # max_seq_len is necessary for Toeplitz.
        expand=2,
        headdim=128,
        device=None,
        dtype=None,
    ):
        factory_kwargs = {"device": device, "dtype": dtype}
        super().__init__()
        self.is_data_dependent = is_data_dependent
        self.d_model = d_model
        self.max_seq_len = max_seq_len
        self.expand = expand
        self.d_inner = self.expand * self.d_model
        self.headdim = headdim
        assert self.d_inner % self.headdim == 0
        self.nheads = self.d_inner // self.headdim

        self.kernel_size = 2 * self.max_seq_len - 1
        self.pad_size = self.max_seq_len - 1
        self.std_dev = 0.5 / np.sqrt(self.max_seq_len)

        if not self.is_data_dependent:
            self.conv_params = nn.Parameter(
                torch.empty(self.nheads, self.kernel_size, dtype=torch.float32, device=
                                                            device)
            )
            nn.init.xavier_uniform_(self.conv_params)

    def forward(self, x, forward_conv=None, reverse_conv=None):
        """
        x: (batch, seqlen, nheads*headdim)
        forward_conv: (batch, seqlen, nheads)
        reverse_conv: (batch, seqlen, nheads)
        """
        residual = x
        x = rearrange(x, 'b l (n h) -> b h n l', n=self.nheads)

        # Pad the hidden states
        x = F.pad(x, (self.pad_size, 0))

        x_fft = torch.fft.fft(x.to(torch.float32), n=2*self.max_seq_len-1)
        if self.is_data_dependent:
            forward_conv = rearrange(forward_conv, 'b l n -> b n l')
            reverse_conv = rearrange(reverse_conv, 'b l n -> b n l')

            conv_params = torch.cat(
                [torch.flip(reverse_conv[:,:,1:], [-1]), forward_conv], dim=-1
            ).to(torch.float32) # FFT requires float32.
            fft_conv_params = torch.fft.fft(conv_params, n=self.kernel_size).unsqueeze(1)
        else:
            fft_conv_params = torch.fft.fft(self.conv_params, n=self.kernel_size)

        output = torch.fft.ifft(x_fft * fft_conv_params, n=self.kernel_size).real
        output = self.std_dev * output[:, :, :, :self.max_seq_len]
        output = rearrange(output, 'b h n l -> b l (n h)').to(residual.dtype) + residual

        return output
```

Figure 7: PyTorch code for the Toeplitz variants in Table 2.

```python
class Vandermonde(nn.Module):
    def __init__(
        self,
        is_data_dependent,
        d_model,
        qk_dim,
        is_dft=True,          # Used only when is_data_dependent is False.
        max_seq_len=None,     # max_seq_len is necessary for non-DFT data-independent
                                                  version.
        expand=2,
        headdim=128,
        device=None,
        dtype=None,
    ):
        factory_kwargs = {"device": device, "dtype": dtype}
        super().__init__()
        self.is_data_dependent = is_data_dependent
        self.d_model = d_model
        self.qk_dim = qk_dim
        self.is_dft = is_dft
        self.max_seq_len = max_seq_len
        self.expand = expand
        self.d_inner = self.expand * self.d_model
        self.headdim = headdim
        assert self.d_inner % self.headdim == 0
        self.nheads = self.d_inner // self.headdim
        self.d_state = self.nheads * qk_dim

        if self.is_data_dependent:
            self.std_dev = 1 / np.sqrt(2 * self.max_seq_len * self.qk_dim)
            self.eps = 1e-3 # Constant to stabilize training.
        else:
            if self.is_dft:
                column_indices = torch.arange(self.max_seq_len)
                row_indices = torch.arange(self.max_seq_len).unsqueeze(1)
                dft_matrix = torch.cos(2 * torch.pi * row_indices * column_indices / self
                                                    .max_seq_len).to(**
                                                    factory_kwargs)
                self.register_buffer('dft_matrix', dft_matrix)
                self.std_dev = 1 / np.sqrt(self.max_seq_len)
            else:
                self.q_bias = nn.Parameter(torch.zeros(self.nheads, self.qk_dim, self.
                                                    max_seq_len, **
                                                    factory_kwargs))
                self.k_bias = nn.Parameter(torch.zeros(self.nheads, self.qk_dim, self.
                                                    max_seq_len, **
                                                    factory_kwargs))
                self.std_dev = 1 / np.sqrt(2 * self.max_seq_len * self.qk_dim)

    def forward(self, v, q=None, k=None):
        batch, seqlen, dim = v.shape
        residual = v
        v = rearrange(v, 'b l (n h) -> b l n h', n=self.nheads)
        if self.is_data_dependent:
            q = rearrange(q, 'b l (n d) -> b n d l', n=self.nheads)
            k = rearrange(k, 'b l (n d) -> b n d l', n=self.nheads)
            q_matrix = torch.cos(
                2 * torch.pi * self.eps * torch.einsum(
                    'b n d t, l -> b n d t l', q, torch.arange(seqlen, dtype=v.dtype).to(
                                                    v.device)
                )
            )
            k_matrix = torch.cos(
                2 * torch.pi * self.eps * torch.einsum(
                    'b n d t, l -> b n d l t', k, torch.arange(seqlen, dtype=v.dtype).to(
                                                    v.device)
                )
            )
            sym_vandermonde = (q_matrix - k_matrix).sum(dim=2)
            output = torch.einsum('b n t l, b l n h -> b t n h', sym_vandermonde, v)
        else:
            if self.is_dft:
                output = torch.einsum('b l n h, t l -> b t n h', v, self.dft_matrix)
            else:
                q, k = self.q_bias, self.k_bias
                q_matrix = torch.cos(
                    2 * torch.pi * torch.einsum(
                        'n d t, l -> n d t l', q, torch.arange(self.max_seq_len, dtype=v.
                                                    dtype).to(v.device)
                    )
                )
                k_matrix = torch.cos(
                    2 * torch.pi * torch.einsum(
                        'n d t, l -> n d l t', k, torch.arange(self.max_seq_len, dtype=v.
                                                    dtype).to(v.device)
                    )
                )
                sym_vandermonde = (q_matrix + k_matrix).sum(dim=1)
                output = torch.einsum('n t l, b t n h -> b t n h', sym_vandermonde, v)
        output = self.std_dev * output
        output = rearrange(output, 'b l n h -> b l (n h)') + residual
        return output
```

Figure 8: PyTorch code for the Vandermonde variants in Table 2.

```python
class Cauchy(nn.Module):
    def __init__(
        self,
        is_data_dependent,
        d_model,
        qk_dim,
        max_seq_len=None,    # max_seq_len is necessary for data-independent version.
        expand=2,
        headdim=128,
        device=None,
        dtype=None,
    ):
        factory_kwargs = {"device": device, "dtype": dtype}
        super().__init__()
        self.is_data_dependent = is_data_dependent
        self.d_model = d_model
        self.qk_dim = qk_dim
        self.max_seq_len = max_seq_len
        self.expand = expand
        self.d_inner = self.expand * self.d_model
        self.headdim = headdim
        assert self.d_inner % self.headdim == 0
        self.nheads = self.d_inner // self.headdim
        self.d_state = self.nheads * qk_dim

        self.tol = 1e-8
        self.std_dev = 1 / np.sqrt(self.max_seq_len * self.qk_dim)
        if self.is_data_dependent:
            self.bias = nn.Parameter(torch.tensor(0.5))
        else:
            self.q_matrix = nn.Parameter(
                torch.empty(self.max_seq_len, self.nheads, self.qk_dim, **factory_kwargs)
            )
            self.k_matrix = nn.Parameter(
                torch.empty(self.max_seq_len, self.nheads, self.qk_dim, **factory_kwargs)
            )
            nn.init.xavier_normal_(self.q_matrix)
            nn.init.xavier_normal_(self.k_matrix)

    def forward(self, v, q=None, k=None):
        residual = v
        v = rearrange(v, 'b l (n h) -> b l n h', n=self.nheads)

        if self.is_data_dependent:
            q = rearrange(q, 'b l (n d) -> b n l 1 d', n=self.nheads)
            k = rearrange(k, 'b l (n d) -> b n 1 l d', n=self.nheads)
            q = torch.exp(q) + self.bias
            k = torch.exp(k) + self.bias

            inv_cauchy_matrix = q + k + self.tol
            cauchy_matrix = torch.sum(1 / inv_cauchy_matrix, dim=-1)

            output = torch.einsum('b t n h, b n l t -> b l n h', v, cauchy_matrix)
        else:
            # q, k: (nheads, seqlen, qkdim)
            q = torch.exp(self.q_matrix)
            k = torch.exp(self.k_matrix)

            inv_cauchy_matrix = (q.unsqueeze(1) + k.unsqueeze(0)) + self.tol
            cauchy_matrix = torch.sum(1 / inv_cauchy_matrix, dim=-1)

            output = torch.einsum('b t n h, l t n -> b l n h', v, cauchy_matrix)

        output = self.std_dev * output
        output = rearrange(output, 'b l n h -> b l (n h)') + residual

        return output
```

Figure 9: PyTorch code for the Cauchy variants in Table 2.

```
class LowRank(nn.Module):
    def __init__(
        self,
        is_data_dependent,
        d_model,
        qk_dim,
        max_seq_len=None,    # max_seq_len is necessary for data-independent version.
        expand=2,
        headdim=128,
        device=None,
        dtype=None,
    ):
        factory_kwargs = {"device": device, "dtype": dtype}
        super().__init__()
        self.is_data_dependent = is_data_dependent
        self.d_model = d_model
        self.qk_dim = qk_dim
        self.max_seq_len = max_seq_len
        self.expand = expand
        self.d_inner = self.expand * self.d_model
        self.headdim = headdim
        assert self.d_inner % self.headdim == 0
        self.nheads = self.d_inner // self.headdim
        self.d_state = self.nheads * qk_dim

        self.std_dev = 1 / np.sqrt(self.max_seq_len * self.qk_dim)
        if not self.is_data_dependent:
            self.q_matrix = nn.Parameter(
                torch.empty(self.max_seq_len, self.nheads, self.qk_dim, **factory_kwargs)
                )
            self.k_matrix = nn.Parameter(
                torch.empty(self.max_seq_len, self.nheads, self.qk_dim, **factory_kwargs)
                )
            nn.init.xavier_normal_(self.q_matrix)
            nn.init.xavier_normal_(self.k_matrix)

    def forward(self, v, q=None, k=None):
        residual = v
        v = rearrange(v, 'b l (n h) -> b l n h', n=self.nheads)

        if self.is_data_dependent:
            q = rearrange(q, 'b l (n d) -> b l n d', n=self.nheads)
            k = rearrange(k, 'b l (n d) -> b l n d', n=self.nheads)
            output = torch.einsum('b t n d, b l n d, b l n h -> b t n h', q, k, v)
        else:
            output = torch.einsum('t n d, l n d, b l n h -> b t n h', self.q_matrix, self
                                    .k_matrix, v)

        output = self.std_dev * output
        output = rearrange(output, 'b l n h -> b l (n h)') + residual

        return output
```

Figure 10: PyTorch code for the low-rank variants in Table 2.

```python
class Attention(nn.Module):
    def __init__(
        self,
        is_data_dependent,
        d_model,
        qk_dim,
        max_seq_len=None,      # max_seq_len is necessary for data-independent version.
        expand=2,
        headdim=128,
        device=None,
        dtype=None,
    ):
        factory_kwargs = {"device": device, "dtype": dtype}
        super().__init__()
        self.is_data_dependent = is_data_dependent
        self.d_model = d_model
        self.qk_dim = qk_dim
        self.max_seq_len = max_seq_len
        self.expand = expand
        self.d_inner = self.expand * self.d_model
        self.headdim = headdim
        assert self.d_inner % self.headdim == 0
        self.nheads = self.d_inner // self.headdim
        self.d_state = self.nheads * qk_dim

        if not self.is_data_dependent:
            self.q_matrix = nn.Parameter(
                torch.empty(self.max_seq_len, self.nheads, self.qk_dim, **factory_kwargs)
                )

            self.k_matrix = nn.Parameter(
                torch.empty(self.max_seq_len, self.nheads, self.qk_dim, **factory_kwargs)
                )

            nn.init.xavier_normal_(self.q_matrix)
            nn.init.xavier_normal_(self.k_matrix)

    def forward(self, v, q=None, k=None):
        residual = v
        v = rearrange(v, 'b l (n h) -> b l n h', n=self.nheads)

        if self.is_data_dependent:
            q = rearrange(q, 'b l (n d) -> b l n d', n=self.nheads)
            k = rearrange(k, 'b l (n d) -> b l n d', n=self.nheads)
            qk = torch.einsum('b t n d, b l n d -> b n t l', q, k)
            attn_weights = torch.softmax(1 / np.sqrt(self.qk_dim) * qk, dim=-1)
            output = torch.einsum('b n t l, b l n h -> b t n h', attn_weights, v)
        else:
            qk = torch.einsum('n t d, n l d -> n t l', self.q_matrix, self.k_matrix)
            attn_weights = torch.softmax(1 / np.sqrt(self.qk_dim) * qk, dim=-1)
            output = torch.einsum('n t l, b l n h -> b t n h', attn_weights, v)

        output = rearrange(output, 'b l n h -> b l (n h)') + residual

        return output
```

Figure 11: PyTorch code for the Attention variants in Table 2.

# G    Background

**FNet**    In the FNet [23] architecture, the sequence mixing module utilizes a Discrete Fourier Transform (DFT) for processing input sequences. One of the approaches they adopt for short sequences is the application of the matrix representation of the DFT to the input sequence. This representation, denoted as $M$, takes the form of a Vandermonde matrix, constructed from the roots of unity:

$$M_{nk} = e^{-\frac{2\pi i}{L} nk} \tag{4}$$

where indices $n$ and $k$ range from 0 to $L-1$. First, we define $\omega_n$ as the $n$-th root of unity:

$$\omega_n = e^{-\frac{2\pi i}{L} n} \tag{5}$$

Then, the matrix $M$, which represents the DFT, is expressed using $\omega_n$ in the form of a Vandermonde matrix as follows:

$$M = \begin{pmatrix} \omega_0^0 & \omega_0^1 & \cdots & \omega_0^{L-1} \\ \omega_1^0 & \omega_1^1 & \cdots & \omega_1^{L-1} \\ \omega_2^0 & \omega_2^1 & \cdots & \omega_2^{L-1} \\ \vdots & \vdots & \ddots & \vdots \\ \omega_{L-1}^0 & \omega_{L-1}^1 & \cdots & \omega_{L-1}^{L-1} \end{pmatrix} \tag{6}$$

**MLP-Mixer**  In the MLP-Mixer architecture, sequence mixing is implemented using a MLP $M$. This MLP operates on each token in the sequence and is formulated as follows:
$$M = W_2 \sigma(W_1)$$
where $\sigma$ denotes a non-linear activation function (such as ReLU). In this context:

- $W_1$ is a weight matrix with dimensions $\mathbb{R}^{d_S \times L}$, transforming the sequence from its original sequence length $L$ to an intermediate dimension $d_S$.
- $W_2$ is a weight matrix with dimensions $\mathbb{R}^{L \times d_S}$, converting the dimensions back from the intermediate dimension $d_S$ to the original sequence length $L$.

This design means that $W_1$ and $W_2$ first compress and then expand the sequence dimensions, respectively. The choice of the inner dimension $d_S$ is made independent of the sequence length $L$. Such a configuration leads to a computational complexity that is linear with respect to the sequence length $L$, contrasting with the quadratic complexity commonly seen in sequence mixing operations in attention mechanisms.

Note that $M$ is a dense matrix, effectively capturing interactions across different tokens in the sequence.

**Linear Attention**  Transformers process sequences of feature vectors, denoted as $\mathbf{X} \in \mathbb{R}^{N \times C}$, where $N$ represents the number of vectors and $C$ their dimension. The core component of a Transformer is the self-attention mechanism, which mixes information across the sequence.

In self-attention, the sequence $\mathbf{X}$ is projected into queries $\mathbf{Q}$, keys $\mathbf{K}$, and values $\mathbf{V}$ using matrices $\mathbf{W}_Q, \mathbf{W}_K, \mathbf{W}_V \in \mathbb{R}^{C \times D}$. The attention output is calculated as follows:

$$\mathbf{Q} = \mathbf{X}\mathbf{W}_Q, \tag{7}$$
$$\mathbf{K} = \mathbf{X}\mathbf{W}_K, \tag{8}$$
$$\mathbf{V} = \mathbf{X}\mathbf{W}_V, \tag{9}$$
$$\text{Attention}(\mathbf{X}) = \text{softmax}\left(\frac{\mathbf{Q}\mathbf{K}^T}{\sqrt{D}}\right)\mathbf{V} \tag{10}$$

A generalized version of self-attention can be represented as [22]:

$$\mathbf{v}'_i = \frac{\sum_{j=1}^{N} \text{similarity}(\mathbf{q}_i, \mathbf{k}_j)\mathbf{v}_j}{\sum_{j=1}^{N} \text{similarity}(\mathbf{q}_i, \mathbf{k}_j)}, \tag{11}$$

where the standard softmax attention employs $\text{similarity}(\mathbf{q}, \mathbf{k}) = e^{\frac{\mathbf{q}^T \mathbf{k}}{\sqrt{D}}}$ as the similarity function.

In the context of multi-headed Linear Attention (LA) [22], the input sequence is preprocessed and projected into three matrices for each head $h \in [H]$. This is achieved through learned parameters and nonlinear transformations. Specifically, for each head, the input is projected into matrices $\sigma(\mathbf{Q}^h)$, $\sigma(\mathbf{K}^h)$, and $\mathbf{V}^h$, where $\sigma$ denotes a non-linearity function. The set of learned parameters for these projections, denoted as $\Theta$, includes $\mathbf{W}_Q, \mathbf{W}_K$ corresponding to the query and key matrices across all heads, defined as $\Theta = (\mathbb{R}^{C \times Hd})^2$. Here, $H$ is the number of heads, and $d$ represents the dimension for each head for keys and queries.

The class of structured matrices $\mathcal{M}$ used in LA is characterized as the class of Low Rank matrices. The matrix-generating function for each head $h$, denoted as $f_{\mathcal{M}}^h$, computes the matrix $\mathbf{M}^h$ as $\mathbf{M}^h = \sigma(\mathbf{Q}^h)\sigma(\mathbf{K}^h)^T$. This computation involves linear projections of the queries and keys, application of element-wise nonlinearity $\sigma$, and matrix multiplication.

The output for each head $h$, denoted as $\mathbf{Y}^h$, is then computed as $\mathbf{Y}^h = \mathbf{M}^h(f_X(\mathbf{X}))^h$, where $f_X = \mathbf{W}_V \in \mathbb{R}^{C \times D}$ represents a projection of the input sequence with $\mathbf{W}_V$ as a learned parameter matrix. This projection transforms the input data $\mathbf{X}$ into a suitable form for processing by the attention mechanism. The LA method, as described, offers computational efficiency with time and memory complexity of $\mathcal{O}(N)$, in contrast to the traditional softmax attention mechanism which scales with $\mathcal{O}(N^2)$.

**Discrete Convolution**  Consider a convolution operation between sequence $X$ and the filter sequence $h$ to produce the output sequence $Y$. Such a convolution can be represented by a matrix multiplication with an $L \times L$ Toeplitz matrix $M$ [31].

Let:

- $X$ be the input signal, a column vector of length $L$, i.e., $X \in \mathbb{R}^{L \times 1}$.

- $h$ be the convolution filter, with a length $N$ where typically $N \leq L$.
- $Y$ be the output of the convolution, also of length $L$.

The Toeplitz matrix $M$ for the convolution is constructed as follows:

$$M = \begin{pmatrix} h_0 & h_{-1} & \cdots & h_{-L+1} \\ h_1 & h_0 & \cdots & h_{-L+2} \\ \vdots & \vdots & \ddots & \vdots \\ h_{L-1} & h_{L-2} & \cdots & h_0 \end{pmatrix}$$

Thus, the output $Y = MX$ represents the convolution of $X$ with $h$, with both $X$ and $Y$ having the same length $L$, achieved through the $L \times L$ Toeplitz matrix $M$ as defined above.

