# OpenReview forum: "Hydra: Bidirectional State Space Models Through Generalized Matrix Mixers"
_NeurIPS.cc/2024/Conference — NeurIPS 2024 poster_

### Official Review · Reviewer_RwkG · 2024-07-09

**Soundness:** 2
**Presentation:** 1
**Contribution:** 3
**Rating:** 5
**Confidence:** 3

**Summary:**

The paper introduces a matrix mixer framework for sequence models which linearly applies an LxL matrix M  to a sequence representation X of length L. Popular sequence models can be framed within this context e.g. softmax self-attention or SSMs, according to different properties of M. The authors use their framework to identify desirable properties of M, such as data dependence or extendability (where the parameterisation is such that the sequence length L can change for different sequences), or efficient matmul. The authors then use their framework to design sequence models with Vandemonde or Cauchy mixing matrices that perform comparably to attention. The main contribution of the paper is the introduction of Hydra, a bi-directional variant of SSM like Mamba. To do this, the authors let M instead denote a quasiseparable matrix (which is a bidirectional variant of the causal semiseparable matrices introduced in the Mamba-2 paper [6]). The authors show that Hydra (i.e. using quasiseparable matrix mixers) outperforms other matrix mixers (4.1), outperforms other approaches for bidirectional SSMs (4.2), and outperform other standard sequence model arhchitectures like transformers on Masked Language Modelling and ImageNet classification.

**Strengths:**

1. The paper provides a general framework for matrix mixing and introduces desirable properties like Sequence Alignment.
2. Introduces a bidirectional version of SSMs, called Hydra, which is motivated through their framework.
3. Hydra seems to work really well, beats unidirectional Mamba and transformers across different settings where bidirectionality is desired.
4. Shows their framework can motivate new sequence models using e.g. Vandermonde or Cauchy matrices, and provides insights into why existing methods may perform well (like low rank matrices in linear attention).

**Weaknesses:**

1. The writing is unclear in large parts. This detracts from the flow and readability, and ultimately makes it somewhat a frustrating read, as it seems there are nice ideas here but communicated poorly. For example:
- What does “common data transformations” mean in definition 2.1.
- The term “Structured matrices” is not standard terminology and is not formally defined. In my reading the closest thing to a definition is line 117: “... a structured matrix, which are known to possess sub-quadratic matrix multiplication”. Is a structured matrix defined by the ability to do sub-quadratic matrix multiplication, because line 117 suggests there are other properties involved. Also the reference to established “mathematical literature” in line 120 but no citations are provided nor examples of structured matrices.
 - “Data dependence” also doesn’t seem to have a theoretical definition, but is part of the theoretical results (proposition 2.3) so should be defined. What does “canonical” mean in the context of line 138? The authors write "Although data-dependency is a popular notion that is widely regarded as being crucial to performance of Attention, it lacked any formal definition in the literature." but there also isn't a formal provided here in my reading. What does "the third interpretation" refer to in line 153?
- What are $\phi$ or $\hat{P}$ in definition 2.2 of sequence alignment, are there any constraints on $\hat{P}$, why/when is it necessary in the definition? I assume $\phi$ is the empty set but this is not obvious nor defined.
- The proof of proposition 2.6 does not prove the sequence alignment property.
- Are Quasiseparable matrices established in the literature or have you defined them? There is no citation provided afaict.
- Should the tilde be a hat on P in line 237 (over vice versa in line 131).
- The 2 paragraphs from line 190 to line 205 make reference to results which are a lot later (e.g. table 2). I would move the table earlier or at least refer to table 2.
- “Taming the Hydra” isn’t particularly insightful for a subsection title.
- The advantages of Hydra (lines 246 to 249) are not justified until later in the subsection, so on first reading it seems like overclaiming (also see my 2nd point). It is also not clear what is being compared to with these advantages (“heuristic alternatives” is cited for the first one but is this also true for the second and third benefit?)
- “It is known that quasiseparable matrices are closed under addition” - cite or prove.
2. It is not clear to me if the theoretical motivation of Quasiseparable matrices is actually a big factor behind the practical gains: in Figure 4 it seems like you are discretising the forward and backward SSD differently, which seems like it will break the symmetry between the A matrix in the forward and backward SSD which is necessary in for the equivalence in proposition 3.2, which to my understanding states in layman terms: “a QS matrix is just two SS matrices that *share parameters*  with flips/shifts”. I could be wrong, but discretise_A is not defined so it is hard to check and the zipped code is different. But if I am right this seems to go against one of the motivations of the paper that previous bidirectional SSMs are heuristic or ad-hoc.
3. Likewise, the arguments in line 250-258 that motivate Hydra as opposed to previous approaches for bidirectional SSMs make it seem like the only difference is the diagonal elements, which can be seen as skip connections (see e.g. https://arxiv.org/abs/2302.10322 in the example of diagonal elements of attention matrices replacing standard skip connections). Given that the previous methods presumably also use skip connections, this difference seems quite minimal. If anything, it might be better to pitch this framework as encapsulating and generalising previous works.
4. Ablations to understand the empirical improvements made in Hydra are missing. In general the results seem great, but as discussed a reader may have unanswered questions why Hydra seems to work so well.  For example, related to above, what to performance happens if you remove the extra diagonal part, which should boil down to the Add method in table 3 right? Also what if you change the convolution (e.g. remove it or make it a causal conv) because it doesn’t seem to be connect to the theory here (of quasiseparable mixers)? Also what happens if you remove the parameter sharing between the two SSDs (which again should get closer to the previous methods)? What if you place the different mixers in a standard transformer block instead of a Mamba block in table 2 (does table 2 give an unfair advantage to QS)?
5. The authors write that "Hardware efficiency" is a limitation in the appendix. Do the authors have throughput results for Hydra compared to transformers? If hardware efficiency is a concern then isn't "fast implementations on hardware" a more practical desiderata to design new sequence models as opposed to use "sub-quadratic matmuls in theory", which is the line taken by this work?

**Questions:**

- Is there a theoretical argument for why data dependence/extendability are nice properties? Or just intuitive? Are there settings where these properties can hurt e.g long range?

Typo:
- linear 38 "they often lack a systematic" - remove "and"

**Limitations:**

There is a discussion of some limitations in the appendix.

---

> ### Author Rebuttal · Authors · 2024-08-07
>
> We sincerely thank the reviewer for recognizing the novelty of our framework, its potential to motivate performant sequence mixers, and superior empirical performance enjoyed by Hydra.
>
> ---
>
> The reviewer's feedback highlights the following key concerns:\
> Q1. Ablating away the shift and the diagonal operations to understand Hydra's performance\
> Q2. Do separate parameters conflict with the motivations of the matrix mixer framework?\
> Q3-1. Clarifications on the definitions of Quasiseparable, and Structured Matrices and Data Dependence, along with further writing style comments.\
> Q3-2. Writing concerns.
>
> A1) We thank the reviewer for this valuable suggestion; the ablation shows that the GLUE scores with only shift or diagonal operations matches the Add variant ($\approx 80.6$). **It is only when both operations are used together do we see an improvement** ($81.7$). This **validates our framework**, which explains this ablation by noting that a Quasiseparable (**QS**) matrix is strictly more expressive than adding two SS matrices.
>
> A2) We refer the reviewer to the definition of QS matrices [C]: *a matrix M is N-QS if its any submatrix from either the strictly upper or lower triangle (off-diagonal) has a rank of at most N.* We remark that the matrix **with non-symmetric upper and lower triangular forms is a general QS matrix** and is consistent with the motivations of the Matrix Mixer framework.
>
> A3-1) We refer the reviewer to [B, C] for the definition of QS matrices. Structured Matrices [A, B, C, 6, 8, 13, 16, 29] are a well-studied area of mathematics, defined as matrices that admit a subquadratic matrix multiplication algorithm. By data dependence in the "third interpretation" we mean that each parameter of a matrix is **either a free parameter or projected from exactly one token**. By canonical data dependence, we mean that the parameters of SAM matrices **can naturally be made data-dependent using the bijective map** $f_{\mathcal{E}}(\cdot)$.
>
> ---
>
> We now provide a detailed response to all of reviewer's comments:
>
> ### A1. Ablations to understand Hydra's performance
>
> We thank the reviewer for suggesting an ablation study to assess the impact of shift and diagonal operations on Hydra's performance. In this study, we removed either the Diagonal or Shift operation, or both (equivalent to the Add method). Our results are tabulated below:
>
> |Method|#Params|$L_{ce}$|Acc (%)|GLUE|
> |-|-|-|-|-|
> |Add|70M|1.68|65.6|80.6|
> |No Diag|70M|1.68| 65.7|*80.7*|
> |No Shift|70M|1.67| 65.8|*80.7*|
> |Quasi|70M|1.66|65.9|**81.7**|
>
> We observe that performance remains unchanged when only one operation is used. **Only when both operations are used together do we see an improvement** ($81.7$ vs $80.6$). This validates our framework, showing that a QS matrix is strictly more expressive than adding two SS matrices.
>
> The reviewer has raised important ablations involving convolution operations and backbone architectures. **While these are beyond the scope of this paper, as we focus on sequence mixers, they are valuable directions for future research.** We refer the reviewer to the second global response where we clarify the focus of our paper on sequence mixers.
>
> ### A2. Do separate parameters conflict the matrix mixer framework?
>
> Yes, the reviewer is correct that the two halves of the matrix are discretized separately. This does not deviate from the motivations of our framework as the matrix **with non-symmetric upper and lower triangular forms is simply a QS matrix**. This follows from the rank characterization of QS matrices [C]: *a matrix $M$ is $N$-QS if any submatrix from either the strictly upper or lower triangle (off-diagonal) has a rank of at most $N$.* We note that this definition applies regardless of parameter sharing between the upper and lower halves, and that the parameter shared matrices referred by the reviewer are a strict subset of QS matrices.
>
> Proposition 3.2 does not include this detail for pedagogical simplicity and it can easily be extended to QS matrices. To see this, divide the QS matrix into strict upper ($SS_u$), strict lower ($SS_l$), and diagonal ($D$) components, and then:
>
> $$QS(X) = \text{shift}(SS_l(X)) + \text{flip}(\text{shift}(SS_u(\text{flip}(X)))) + DX$$
>
> ### A3-2. Writing concerns
> - “Common data transformations” refers to standard projects commonly used in the ML community, such as those implemented using convolutions or linear layers.
> - Components in Definition 2.2 are necessary as they allow for multiple sets of parameters like B and C. As assumed, $\emptyset$ represents the empty set.
> - By the definition of SAM, Linear Attention is sequence aligned as each element $(i,j)$ in the matrix $m_{ij}$ is parameterized by $Q_i$ and $K_j$.
> - Quasiseparable matrix are closed under addition [B]
> - Efficacy of data dependence and extendability is born out of the experience of the community and has been shown to be effective on long range tasks [6, 16]
>
> We appreciate the reviewers constructive feedback on the writing, such as the typo in Line 237 and the ordering of the paper. We will address all the aforementioned issues in our next revision.
>
> ### A4. Is Hydra hardware efficient?
>
> We agree with the reviewer that hardware efficiency is indeed an important desideratum for designing new sequence models. However, achieving hardware efficiency involves a series of complex factors, making it challenging to directly target hardware-friendly models from the outset. Therefore, computational complexity is usually prioritized first, with subsequent efforts focused on developing efficient implementations.
>
> Hydra leverages hardware-friendly kernels from Mamba, maintaining competitive speed with Transformers for short sequences. As sequence lengths increase, Hydra’s sub-quadratic design allows it to surpass Transformers in speed, addressing the quadratic bottleneck. The growing interest in processing long sequences [D, E, F] further underscores the importance of Hydra.

---

> ### Author Response · Authors · 2024-08-07
> **References**
>
> [A] Xia, Jianlin, et al. “Fast algorithms for hierarchically semiseparable matrices.” Numerical Linear Algebra with Applications, 17.6 (2010): 953-976.\
> [B] Boito, Paola. “Matrix Structures and Matrix Functions.” Proceedings of the 2023 International Symposium on Symbolic and Algebraic Computation. 2023.\
> [C] Pernet, Clément, Hippolyte Signargout, and Gilles Villard. "Exact computations with quasiseparable matrices." Proceedings of the 2023 International Symposium on Symbolic and Algebraic Computation. 2023.\
> [D] Bertsch, Amanda, et al. "Unlimiformer: Long-range transformers with unlimited length input." Advances in Neural Information Processing Systems 36. 2023.\
> [E] Ding, Jiayu, et al. "Longnet: Scaling transformers to 1,000,000,000 tokens." arXiv preprint arXiv:2307.02486. 2023.\
> [F] Liu, Hao, Matei Zaharia, and Pieter Abbeel. "Ring attention with blockwise transformers for near-infinite context." International Conference on Learning Representations. 2024.

---

> > ### Comment · Reviewer_RwkG · 2024-08-12
> > **Thanks for the response**
> >
> > Thank you for the clarifications, and ablation on diagonal/shift operations. My concerns are largely addressed, although I view the further ablations on the convolution and block structure as within the scope of the submission and would encourage the authors to perform these ablations. Thanks in advance also for updating the presentation in the next revision, though the promised updates to the writing represent significant changes to the original submission's presentation which make it hard to assess the original submission.
> >
> > I have an additional question following the diagonal/shift ablation:
> > - Does the "No shift" row in the ablation have the same expressivity as a QS matrix in the definition of [C]? The diagonal terms should be the sum of the $SS_l$ and $SS_u$ diagonal terms along with the $DX$ term, which should have the same expressivity as the $DX$ term I believe.

---

> ### Author Response · Authors · 2024-08-12
>
> We are grateful to the reviewer for their continued engagement and thoughtful questions. And we appreciate this opportunity to provide further clarification on their concerns:
>
> ---
>
> ### 1. Expressivity of the "No Shift/Add" ([Figure, c](https://photos.app.goo.gl/xAsegE2NcLAwrMi96)) variant :
>
> We would first like to clarify that there is no data-dependent $DX$ term in the "No Shift" variant. Intuitively, the "No Shift" variant exhibits reduced expressivity compared to general QS matrices ([Figure, d](https://photos.app.goo.gl/xAsegE2NcLAwrMi96)) because the diagonals in $SS_l$ and $SS_u$, specifically $\overrightarrow{c}^T \overrightarrow{b} + \overleftarrow{c}^T \overleftarrow{b}$, share parameters with the non-diagonal elements. This **parameter tying** limits the model’s expressivity, as *it cannot independently assign values to the diagonal elements*.
>
> Mathematically, observe that the "No Shift" variant satisfies the definition of an N-QS matrix [C] and therefore is a **subset** of the N-QS matrices ([Figure, d](https://photos.app.goo.gl/xAsegE2NcLAwrMi96)). Furthermore, the "No Shift" variant is a **strict subset** of general N-QS matrices because the lack of shift forces additional constraints on the matrix class. For instance, in general QS matrices, the diagonal is freely parameterized, while in the "No Shift" variant, it is a function of non-diagonal elements, $\overrightarrow{c}, \overrightarrow{b}, \overleftarrow{c}, \overleftarrow{b}$.
>
> Our additional ablations reported in our first response (A.1) at the reviewer's suggestion validate this increase in expressivity by demonstrating an improved performance of N-QS matrices compared to the "No Shift" and the "No Diagonal" variants, which are strict subsets of N-QS matrices.
>
> ---
>
> ### 2. Role of Convolution and Block Structure
>
> We would like to begin by noting that **a short convolution within the block is a standard element** used across a vast majority of sub-quadratic models like Mamba2[6], H3[14], Mamba[16], Hyena[29]. We fully agree with the reviewer's insight that investigating the impact of block structure on the performance of the matrix mixer is important, and this is indeed being explored in parallel works like [G]. However, in our work, we focused specifically on **fixing a sub-quadratic block [6] to control for the impact of the matrix mixer**. We simply selected the block architecture from Mamba2 [6] as it is one of the latest iterations of sub-quadratic models. With this view, we sincerely believe that the reviewer’s suggestion of understanding the impact of these architectural choices outside of the matrix mixer would be an important follow-up work.
>
> ---
>
> ### 3. Scope of Changes
>
> We are truly grateful for the reviewer's valuable suggestions to improve the clarity and accessibility of our work. However, we would like to emphasize that our changes primarily involve adding more precise and mathematically complete definitions, and this would not alter the core results or contributions of the paper.
>
> ---
>
>
> [G] Yang, Songlin, et al. "Parallelizing Linear Transformers with the Delta Rule over Sequence Length." arXiv preprint arXiv:2406.06484 (2024).

---

> > ### Comment · Reviewer_RwkG · 2024-08-13
> > **Thanks again**
> >
> > Thanks again for the discussion.
> >
> > Regarding 1., thank you for the response and figure. My understanding of the "No shift" ablation to Hydra was that only the shift would be removed and you would still have the data-dependent diagonal component of Hydra. More concretely, you would have the data-dependent diagonal $\delta$ added to Figure c in the linked figure, which should have the same expressivity because the $\delta$s are free to "unlearn" the existing diagonals in Figure c. Is that correct or have I missed something?
> >
> > Regarding 2. and 3., I see where the authors are coming from and don't have major disagreements, but will clarify my points of view which differ from that of the authors. Regarding 2., I maintain that these ablations would strengthen the current work, and encourage the authors to do so. Different architectural components interact in complicated ways, and it would make the matrix mixers proposed here (like Hydra) much more compelling in my view if they can perform well across different backbones. Regarding 3., the accessibility of the contributions to readers is very important in my view, alongside the contributions themselves.

---

> ### Author Response · Authors · 2024-08-14
> **Thank you for your response**
>
> We genuinely appreciate and thank the reviewer for their continued interest and detailed consideration of our work.
>
> ### 1. Expressivity of QS ablation matrices
>
>
> We would like to clarify our previous response: we commented on the expressivity of ([Figure, c](https://photos.app.goo.gl/xAsegE2NcLAwrMi96)) which corresponds to the "Add" ablation and not the "No-Shift" ablation. We apologize for the misunderstanding and would like to take the opportunity to clarify the expressivity of all variants of the quasiseparable ablations that the reviewer suggested.
>
> For convenience, we reproduce the ablation table below:
>
> | Method   | #Params | $L_{ce}$ | Acc (%) | GLUE |
> |----------|---------|----------|---------|------|
> | Add      | 70M     | 1.68     | 65.6    | 80.6 |
> | No Diag  | 70M     | 1.68     | 65.7    | *80.7* |
> | No Shift | 70M     | *1.67*     | *65.8*    | *80.7* |
> | QS    | 70M     | **1.66**     | **65.9**    | **81.7** |
>
> ---
> ---
>
> **Lemma:** The matrix classes satisfy the following inclusions:
> $$\text{Add} \subseteq \text{No-Shift} \subseteq \text{QS}, \quad \text{and} \quad \text{No-Diag} \subseteq \text{QS}.$$
>
> **Proof:**
> 1. ### $\text{Add} \subseteq \text{No-Shift}$
>
> To see this, observe that we can characterize the set of No-Shift matrices as:
> $$ \text{No-Shift} = \\{ W + D \\: | \\: \forall W \in \text{Add}, \\: \forall D \in \text{diag}(\mathbb{R}^d)\\}.$$Choosing $D$ to be the zero matrix, we have the result:
> $$ \text{No-Shift} \supseteq  \\{ W + \mathbf{0} \\: | \\: \forall W \in \text{Add}\\} = \text{Add}$$
>
> ---
>
> 2. ### $\text{No-Shift} \subseteq \text{QS}$
>
> Let $\tilde{SS}_u$ and $\tilde{SS}_l$ denote the (the tilde represents "not strict") upper-triangular and lower-triangular N-Semiseparable matrices. Then,
> $$ \text{No-Shift} = \\{ M = \tilde{SS}_u + \tilde{SS}_l + D \\: | \\: \forall \tilde{SS}_u,\\: \tilde{SS}_l, \\: \forall D \in \text{diag}(\mathbb{R}^d)\\}.$$ We apply the definition of N-QS matrices [C]. Without loss of generality, consider any submatrix $L$ in the strict lower triangular half of $M$. Observe that $L$ is also a submatrix of $\tilde{SS}_l$. By the definition of N-Semiseparable matrices, $\text{Rank}(L) \le \text{N}$, which implies $M \in \text{QS}$, and hence $\text{No-Shift} \subseteq \text{QS}$
>
> We also kindly refer the reviewer to [H, §4.1.5], which discusses this result in detail.
>
> ---
>
> 3. ### $\text{No-Diag} \subseteq \text{QS}$
>
> Let $SS_u$ and $SS_l$ denote strict upper-triangular and strict lower-triangular parts of a N-Quasiseparable matrices. Then,
> $$ \text{No-Diag} =  \\{ SS_u + SS_l \\: | \\: \forall SS_u,SS_l \\},$$
>
> $$ \text{QS} =  \\{ SS_u + SS_l + D\\: | \\: \forall SS_u,SS_l, \\: \\: \forall D \in \text{diag}(\mathbb{R^d}) \\}.$$
>
> Restricting $D$ to be the zero matrix in the definition of $\text{QS}$, we have the required result.
>
> ---
> ---
>
> We now tie the expressivity of different variants with their empirical performance: higher the expressivity, higher the GLUE score. That is the performance follows the same order: $\text{QS}(81.7) > \text{No-Shift}(80.7) > \text{Add}(80.6)$, and that $\text{QS}(81.7) > \text{No-Diag}(80.7)$. This validates our premise of principally deriving QS from the Matrix Mixer framework over using its heuristical counterparts.
>
> We again thank the reviewer for suggesting this ablation. We have now added it to our paper and incorporated our discussions on expressivity into the appendix.
>
> ---
> [H]: Eidelman, Y., Gohberg, I., & Haimovici, I. (2013). *Separable Type Representations of Matrices and Fast Algorithms: Volume 1 Basics. Completion Problems. Multiplication and Inversion Algorithms*. Springer, Basel, Switzerland. [Link](https://link.springer.com/10.1007/978-3-0348-0606-0).

---

> ### Author Response · Authors · 2024-08-14
>
> ### Further ablations on the impact of convolution
>
> We appreciate the reviewer's interest in understanding the impact of non-matrix mixer backbone components, such as convolution, on the model's performance. Below, we provide an ablation study to quantify the impact of removing the short convolution from the backbone. Due to the time constraints, we are currently comparing the variants on their pretraining validation accuracy and cross-entropy loss.
>
>
> | Method | #Params | Conv | $L_{ce}$ | Acc (%) |
> | ------ | ------- | ---- | -------- | ------- |
> | Add    | 70M     | ❌   | 1.70     | 65.3    |
> | Quasi  | 70M     | ❌   | *1.68*     | *65.6*    |
> | Add    | 70M     | ✅   | *1.68*     | *65.6*    |
> | Quasi  | 70M     | ✅   | **1.66**     | **65.9**    |
>
> We make the following observations: 1) QS consistently outperforms the Add variant, both with and without convolution. 2) Adding a short convolution consistently improves the performance of both Add and QS.
>
> This indicates that the established advantages of Quasi over heuristics like Add, Concat, and Multiply persist across changes to the underlying backbone. At the same time, we fully agree with the reviewer that further studies on backbone components are valuable and can indeed help enhance the performance of sequence mixers in general.

---

> > ### Comment · Reviewer_RwkG · 2024-08-14
> > **Author response**
> >
> > I thank the authors for their efforts in responding to my questions.
> >
> > ## No-Shift
> > The response has not clarified my question. In the previous reply, the authors stated that "We would first like to clarify that there is no data-dependent $DX$ term in the No Shift variant". But now in the Lemma the $DX$ term appears in the "No-shift" variant. Does the No-Shift result in the ablation table have the $DX$ term included or not?
> >
> > Moreover, in the Lemma (part 2) you only prove $\text{No-Shift} \subseteq \text{QS}$ but do not show that there is a strict subset. I maintain that the expressivity is the same so that the Lemma part 2 is an equality; please correct me if I am wrong. But if so then the expressivity argument you have made for the ablation table doesn't hold. I agree that the Lemma would make a nice addition to the paper, but only if there is a clear and consistent message for the readers to take away.
> >
> > ## Convolution
> > Thank you for this ablation. I would include this to strengthen the arguments of the paper.

---

### Official Review · Reviewer_5fYM · 2024-07-14

**Soundness:** 4
**Presentation:** 4
**Contribution:** 4
**Rating:** 9
**Confidence:** 4

**Summary:**

This paper presents Hydra, an innovative framework that builds upon the Mamba model by introducing bidirectional capabilities. Hydra's approach centers on a matrix mixer perspective, which allows it to consolidate various sequence models, including Transformers and structured state space models, into a unified framework. The primary strength of Hydra is its capacity to surpass other sequence models in non-causal tasks while retaining efficiency and expressiveness. The study demonstrates how Hydra's novel bidirectional methodology and matrix parameterizations effectively enhance the performance of sequence models.

**Strengths:**

- The authors proposed a novel framework, Hydra, which extends mamba with bidirectional capabilities and presents an interesting perspective on improving sequence models.
- The proposed method of matrix mix offers a cohesive understanding of various sequence model architectures, which also offers valuable insights into how matrix parameterizations and sequence alignment affect model flexibility and performance.
- The paper is well-structured, making it easy for readers to follow the development of the Hydra framework and its contributions to sequence modeling.
- Abundant experiments covering both language and vision tasks illustrate the efficacy of the proposed method.

**Weaknesses:**

I don't identify a specific weakness, but I have a few questions regarding efficiency. Please see the following section.

**Questions:**

- The experiments are mainly around ~ a 100 M parameter scale. I am curious about whether the model can be scaled up to ~1.5B parameters and how the model will perform.
- I am wondering, in such non-causal tasks, how is a comparison between the proposed hydra framework with stacked bidirectional Mamba frameworks (https://arxiv.org/pdf/2401.09417,https://arxiv.org/abs/2404.15772)?
- Compared to the original Mamba framework, does Hydra require additional computational resources and training time, as the training objective is harder? I am also curious about whether the matrix mix method will harm performance on general tasks.

**Limitations:**

See Questions.

---

> ### Author Rebuttal · Authors · 2024-08-06
>
> We sincerely thank the reviewer for the thorough evaluation and for recognizing the novel contributions of our framework. We are pleased that the reviewer appreciated the **bidirectional capabilities**, the **valuable insights provided by the matrix mixer perspective**, and the **extensive experimental validation** across both language and vision tasks.
>
> ---
>
> We now address the reviewer’s key questions:
>
> ### Scalability to larger models (~1.5B parameters)
>
> While our experiments primarily focused on models with ~100M parameters due to resource constraints, Hydra is certainly scalable. We anticipate that Hydra will continue to demonstrate effectiveness in non-causal tasks as the model size increases.
>
> ### Comparison with stacked bidirectional mamba frameworks:
>
> We appreciate the suggestion to compare Hydra with previous stacked bidirectional Mamba frameworks (e.g., provided references). Under the matrix mixer perspective, most of **these methods fall into Add, Mult, or Concat categories** listed in Table 3. As demonstrated, the quasiseparable matrix mixer used in **Hydra outperforms other bidirectional variants.** Therefore, we confidently expect that substituting the bidirectional components of previous models with Hydra would lead to a boost in performance.
>
> ### Computational resources and training time:
>
> As similar to all other bidirectional extensions of Mamba [12, 15, 42], Hydra does introduce additional computations due to its bidirectional nature. However, unlike many of the previous extensions that utilize two completely separate unidirectional components, Hydra greatly reduces this overhead by sharing the projection layer for both forward and backward sequence processing and by batching both directions to maximize GPU utilization. This approach results in only a ~30% reduction in throughput compared to Mamba.
>
> ### Performance of different matrix mixers on general tasks:
>
> Hydra, derived from the matrix mixer framework, has demonstrated significantly stable training and enhanced performance across well-established domains such as vision and NLP. We are confident that models developed using the matrix mixer framework, when logically designed for specific domains, will continue to achieve superior performance across a diverse range of tasks.

---

> > ### Comment · Reviewer_5fYM · 2024-08-07
> > **Acknowledgement**
> >
> > Thank you for your response. Most of my concerns have been addressed and I will keep my very positive score. I appreciate the significant contribution and merit of this work as an important follow-up to mamba.

---

> > > ### Author Response · Authors · 2024-08-12
> > > **Statement of Thanks**
> > >
> > > We sincerely thank the reviewer for their very positive feedback and for taking the time to engage with our work. We are glad that we have addressed all their concerns and we appreciate their recognition of the significance and merit of our work.

---

### Official Review · Reviewer_onHY · 2024-07-14

**Soundness:** 3
**Presentation:** 2
**Contribution:** 3
**Rating:** 6
**Confidence:** 4

**Summary:**

The paper introduces the concept of matrix mixer and sequence alignment for explaining recent sequence models including Transformer, linear transformer, and Mamba. It also proposes a quasiseparable matrix mixer (Hydra) as an alternative to the bidirectional SSM model. The experiments show that the quasiseparable matrix structure performs better than others including low rank (linear attention), attention, and dense. Also, Hydra outperforms other naive bidirectional approaches and some baselines for masked language modeling and image classification.

**Strengths:**

1. The idea of designing a structure matrix for a bidirectional case is reasonable and effective while the implementation is simple.
2. The explanation of different matrix mixers, their relation to other methods, and the advantages/disadvantages are clear. This is very informative.
3. The ablation study shows how different matrix mixers perform and the benefit of quasiseparable matrix structure.

**Weaknesses:**

1. The presentation of the paper needs to be improved.
    1. It contains unnecessary details. The main contributions of the paper are matrix mixer, sequence alignment, and Hydra, but the focus diverges throughout the paper, especially in Section 2. Sections 2.3 and 2.4 can be moved as a side note after introducing Hydra.
    2. The purpose of introducing Cauchy and Vandermonde matrix mixers is unclear. How can this be useful or helpful in some ways? It's not clearly explained or shown in the experiments.

Overall, the current representation makes it difficult for the readers to understand the core contributions.

2. The experiments shown in the paper are limited. The main comparison table (Table 4) does not include any recent transformers or mamba-based models. Also, the experiments include one example of non-causal language modeling (masked language modeling) and image classification as applications for Hydra (bidirectional model settings). Could authors show at least one more example application that Hydra can be useful and better than other comparable models?

**Questions:**

1. Is there any speed benefit with the quasiseparable bidirectional model compared to the naive bidirectional approaches? It would be good to include some speed comparisons in Table 3.

**Limitations:**

The limitations are discussed and are reasonable.

---

> ### Author Rebuttal · Authors · 2024-08-07
>
> We thank the reviewer for their insightful comments and we are glad that they **found our Matrix Mixer framework informative and the Hydra model simple and effective**.
>
> The review focuses on the following concerns:\
> Q1. The main comparison table (Table 4) does not include any recent transformers or mamba-based models. Could authors show at least one more example application that Hydra can be useful?\
> Q2. Paper presentation issues\
> Q3. Speed comparisons between Quasiseparable and other naive approaches
>
> ---
> We now answer these questions in detail below:
>
> ### A1. Table 4 does not include recent models and the two domains chosen are limited
>
> > (Table 4) does not include any recent transformers or mamba-based models
>
> We reiterate that the primary objective of Table 4 is to compare the core **sequence mixers** proposed by earlier works. Many recent works, including Caduceus [36], BiGS [42], and Vision Mamba [46], simply **use the Add, Concat, or Element-wise multiplication heuristic on Mamba. These comparisons have already been included in our experiments** in Table 3.
>
> We acknowledge that integrating the Quasiseparable matrix mixer into domain-specific backbones presents excellent directions for domain-specific future work; however, these efforts are beyond the scope of this paper.
>
> > Could authors show at least one more example application that Hydra can be useful?
>
> We chose the MLM and ImageNet-1k classification tasks since **they are canonical tasks used by previous works that proposed new sequence mixers like M2[13] and Hyena[29] and their baselines are well-tuned and established**. We are confident our method will also excel in other domains such as DNA modeling and graphs. We invite domain-specific research communities to explore Quasiseparable in their applications.
>
> ---
>
> ### A2. Paper presentation issues
>
> We appreciate the reviewer’s feedback on the structure of our paper. We would like to convey that we structured our paper this way to match the flow of our contributions:
>
> 1. **Formalization of the Sequence Mixer:** In Section 2.1, we formalize a sequence mixer as a generalized matrix mixer, identifying the computational bottleneck as the cost associated with the matrix class. This leads us to focus on structured matrices.
> 2. **Optimal Structured Matrices:** In Section 2.2, we define SAM matrices, characterizing structured matrices that enjoy data dependence and extendability to determine which classes would have superior empirical performance.
> 3. **Framework Generality:** In Section 2.3, we demonstrate the broad applicability of our framework by showing that more than 13 past methods, including Attention [40], Mamba [6, 16], MLP-Mixer [38], and Linear Attention [22], fall under this paradigm, indicating our framework’s generality.
> 4. **Validation and Prescriptive Power:** In Section 2.4, we validate our framework's prescriptions by identifying Cauchy, Vandermonde, and Quasiseparable matrices as Sequence Aligned Matrices (SAM) which have not been fully utilized in past works and showing that they exhibit strong empirical performance.
> 5. **Hydra:** We chose Quasiseparable matrices due to their hardware properties and connections to Mamba, scaling them up and proposing them as the next sequence mixer of choice. We show that Quasiseparable outperforms other sequence mixers on bidirectional language modeling (+0.8 over BERT) and ImageNet-1k image classification (+2.2 over ViT).
>
> We request the reviewer to also take a look at the first global response wherein we enlist our core contributions in depth and tie them to the different sections of the paper
>
> ---
>
> ### A3. Speed comparison between QS and naive approaches
>
> Hydra shares the projection layers for both forward and backward passes, with the additional computation over Mamba being the backward SSD. However, the computations for forward and backward passes can be parallelized by batching, resulting in only a ~30% reduction in throughput compared to Mamba.

---

> > ### Comment · Reviewer_onHY · 2024-08-13
> > **response to the rebuttal**
> >
> > Thank you for the answers.
> >
> > I understand that the paper focuses on introducing the matrix mixer framework and Hydra as an application of the Quasiseparable Matrix (the best matrix mixer). Therefore, the authors think my suggestions about adding recent SOTA comparisons and including another application for Hydra are out of scope. I believe that improving the experiments provides a better understanding of the framework (quality) and the practical use of Hydra. This will enhance the overall quality of the paper.
> >
> > Also, regarding my concern about the presentation of the paper, the authors believe that the current structure better matches the flow of the contributions. I found that the current flow distracts from understanding the core ideas (the reviewer RwkG09 also pointed out). It's because the paper introduces many terms/methods, but the connections between them are unclear. This makes the paper difficult to read. I suggest simplifying the paper, so the reliability can improve.
> >
> > Overall, the paper introduces a novel framework with great insight. However, due to the weaknesses mentioned above, I retain my rating.

---

### Official Review · Reviewer_9VST · 2024-07-16

**Soundness:** 3
**Presentation:** 3
**Contribution:** 2
**Rating:** 4
**Confidence:** 4

**Summary:**

Most of sequence models include the token mixers and the channel mixers, and this paper provides a detailed summary. They also identify the matrix parameterization is crucial for recent SSMs. Therefore, they extend the Mamba model by adding bidirectional quasiseparable matrix mixer. The experiments on GLUE benchmark and ImageNet data verify their method.

**Strengths:**

1. This paper summarizes the previous relevant methods very well, as illustrated in Table 1. The authors also claim that the following two properties are important for sequence aligned matrices: data dependency and extendability. The former one is a well-known propoerty.

2. The proposed quasiseparable matrix mixer is simple and easy to understand. It is implemented through two Semiseparable matrixes.

3. The provided results show the advancement of their method.

**Weaknesses:**

1. The authors claim that the sequence model should be extended beyond their trained length, but they didn't provide the corresponding results.

2. I am curious about the computational complexity of Hydra compared with Mamba. Because they use two SSD operations as shown in Figure 4. The ablation studies are also necessary for the different variants.

3. Please compare with the most advanced methods, such as xLSTM. It also expands the hidden state into a matrix form.

**Questions:**

1. The methods to implement token mixer include not only structured matrix mixer, such as implicit reparameterization of CKConv. So the conclusion in Line 117 is inaccurate. What do you think are the advantages and disadvantages of implicit matrices over structured matrices?

2. How to understand "Sequence Aligned" property in Table 1?

---

> ### Author Rebuttal · Authors · 2024-08-07
>
> We thank the reviewer for recognizing the **detailed categorization of prior methods using the Matrix Mixer framework and for appreciating Hydra’s simple yet effective design**.
>
> ---
>
> We now address the reviewer’s concerns as follows:
>
> ### A1. Ablations on extendability:
> The extendable property of sequence mixers is implicitly validated throughout our experimental section. Non-SAM matrix mixers, such as the Dense variant (e.g., MLP-Mixer [38]), lack extendability and require retraining from scratch when adapting to different sequence lengths. Therefore, in Table 2, we fixed the sequence length to 128 across all variants to ensure a fair comparison, thus emphasizing data dependence (DD) only. Conversely, Hydra, as a SAM matrix mixer, inherently supports both data dependence and extendability. Specifically, in Table 4, Hydra was pretrained on sequences of length 128 (C4) and then fine-tuned with sequences of length 256 (GLUE).
>
> ### A2. Computational complexity of Hydra compared to Mamba:
> Hydra shares the projection layers for both forward and backward passes, with the additional computation over Mamba being the backward SSD. However, the computations for forward and backward passes can be parallelized by batching, resulting in only a ~30% reduction in throughput compared to Mamba.
>
> ### A3. Comparisons to Advanced Methods such as xLSTM
> We thank the reviewer for pointing out xLSTM, and indeed there are many recent subquadratic models that have shown to be performant. We specifically chose to focus on Mamba as it is a recent model that introduces an interpretation of recurrent models as **matrix mixers**, which is the main focus of our work. Furthermore, **since xLSTM is language model and does not have canonical bidriectional variants so it is not a suitable for comparison on bidirectional tasks**.
>
>
> ### A4. Analysis on Implicit Reparameterization:
> We appreciate the reviewer’s observation that the same matrix class can be parameterized differently. However, this property is orthogonal to the underlying matrix class and its computational complexity. **Specifically, an implicitly parameterized mixer like CKConv [35] and S4 [18] is indeed a Toeplitz matrix mixer, as indicated in Table 1**. Therefore, the conclusion in Line 117 remains accurate.
>
>
> ### A5. Clarification of the ‘Sequence Aligned’ Terminology in Table 1:
> The formal definition of Sequence Aligned Matrices (SAMs) is provided in Definition 2.2. To summarize informally, SAMs ensure that the parameters for every submatrix $M[: i+1, :i+1]$ are functions only of the tokens up to index $i$.

---

### Author Rebuttal · Authors · 2024-08-06

# Global Response

---

We express our sincere gratitude to the reviewers for their valuable feedback and constructive suggestions. We are glad that they found our Matrix Mixer framework insightful and informative, and that they appreciated Hydra's simplicity and strong empirical performance.

In this shared response, **we aim to contextualize our paper within the existing literature** by highlighting our core contributions and outlining the scope and impact of our findings. We wish to emphasize that the **Matrix Mixer (MM) framework is a robust tool for creating performant sequence mixers**, as evidenced by the new sequence mixers like Quasiseparable (QS), Cauchy, and Vandermonde developed using MM prescriptions. We also note that these **mixers can seamlessly integrate with various domain-specific backbones** developed by the community. Our goal is to **showcase that this framework is effective and to advocate for its adoption in developing new sequence mixers.**

---

## Core contributions

### 1. The Matrix Mixer Framework

Unlike traditional paradigms that assess a method in its entirety with all its bells and whistles, our **formal** framework offers a different perspective to study models by focusing on their associated matrix mixer which is the core component of a sequence mixer.

We demonstrate that this framework is effective as it possesses the following desirable properties:

1.1. **A formal treatment**

- We formally define a sequence mixer as a **generalized matrix mixer** and identify the computational bottleneck as the cost associated with the matrix class. This prompts us to **focus on structured matrices** that enjoy a fast matrix multiplication algorithm.
- We then seek to determine whether some structured matrices could exhibit superior empirical performance than others. It is widely acknowledged in the machine learning community that data dependency is crucial for performance, and extendability is practically useful. Using these as our desiderata, **our formal framework allows us to define Sequence Aligned Matrices (SAM)**, which characterize matrices possessing these properties.

1.2. **Broad applicability across past methods**
- Next, we demonstrate that our framework is **highly general** and capable of representing a wide range of previous works. In Section 2.3, we show that **more than 13 past methods**, including Attention [40], Mamba [6, 16], MLP-Mixer [38], and LA [22], can be subsumed under our paradigm.
- Moreover, this broad applicability allows us to **compare these previous works on an equal footing**, as shown in Tables 1 and 2, eliminating extraneous complexities of the backbone model architecture.

1.3. **The framework provides effective prescriptions**

- To **validate our framework and test its predictions**, in Section 2.4, we identify Vandermonde, Cauchy, and QS matrices as SAM matrices which have not been fully utilized in past works. Our experiments demonstrate that all these matrices exhibit strong empirical performance, with **QS outperforming all others** and **Cauchy being competitive with Low Rank.**
- Another important validation of our framework is our success with Vandermonde matrices outperforming DFT matrices (a special case of Vandermonde matrices). **FNet [23] (Appendix A.3) previously attempted and failed to make DFT matrices data-dependent and performant.**

This demonstrates that our framework is not only descriptive but also prescriptive, which is a hallmark of robust generalization.

### 2. Hydra: A comprehensive validation and application of QS matrices

- Our framework and ablations suggest that **QS matrices have the potential to be the sequence mixer of choice** for the next generation of sub-quadratic sequence mixers. This potential is further accentuated by the fact that in addition to strong empirical performance, QS matrices also possess the following desirable properties:
  - *Connections to Mamba*: Since QS matrices generalize Semiseparable matrices, they are a **natural mathematical extension of Mamba** to bidirectional settings.
  - *Hardware Efficiency*: QS matrices can be implemented with Mamba as a subroutine, **enabling the use of Mamba's hardware-efficient Triton kernel.**

  To substantiate our claim, we introduced **Hydra**, a bidirectional model with QS matrix as its sequence mixer. Our evaluations on two canonical domains: language and images demonstrated the superior performance of **Hydra over BERT (110M) and ViT-B (88M), achieving +0.8 and +2.2 improvements, respectively.**

- Hydra also addresses an ongoing research question in the ML community: **how to make Mamba bidirectional**. In response, numerous methods, including Caduceus [36], BiGS [42], Vision Mamba [46], and MH-SSM [12], have proposed solutions that, when viewed under our framework, employ one of three heuristics: Add, Concat, or Element-wise multiplication. Our results demonstrate that our method **outperforms all these heuristic approaches** (see Table 3). We provide a new ablation study to support our argument (see Response A1 for Reviewer RwkG).

---

## Clarifying the Focus: Sequence Mixer vs. Model Architecture

Our primary objective is to develop a framework that enables us to **derive performant sub-quadratic sequence mixer alternatives** to attention. Consequently, our methodology focuses on the sequence mixers and **does not address the backbone model architecture**, which is typically tuned to meet domain-specific requirements. Therefore,the appropriate comparisons are with other sequence mixers, including attention. For instance, in vision, ViT [11] is a standard backbone that employs attention, whereas Swin Transformer [24], which adopts a hierarchical structure suitable for images, also uses attention.

We would like to emphasize that these two orthogonal **developments actually complement each other**. We invite domain-specific research communities to consider QS as a potential sequence mixer in their respective domains.

---

### Decision · Program_Chairs · 2024-09-25

**Decision:**

Accept (poster)

**Comment:**

The paper presents a compelling formal framework called the Matrix Mixer Framework, which effectively extends sequence models, including Transformers and SSMs, by incorporating a bidirectional quasiseparable matrix mixer. The introduction of Hydra, a novel extension of the Mamba model, showcases significant advancements in non-causal tasks like masked language modeling and image classification. The paper’s contribution is substantial, demonstrating superior performance over existing approaches, particularly through its strategic use of matrix parameterization and bidirectional capabilities. Nevertheless, the paper’s presentation could be improved for clarity and readability. Despite this, I recommend accepting the paper given the strength of its methodology and results.